# Transcriptional and spatial profiling of the kidney allograft unravels a central role for FcγRIII+ innate immune cells in rejection

Baptiste Lamarthée [1,2,21], Jasper Callemeyn[1,3,21], Yannick Van Herck[4,21], Asier Antoranz[5,21], Dany Anglicheau [6,7], Patrick Boada[8], Jan Ulrich Becker[9], Tim Debyser[1], Frederik De Smet [5], Katrien De Vusser[1,3], Maëva Eloudzeri[7], Amelie Franken[10,11], Wilfried Gwinner [12], Priyanka Koshy [13], Dirk Kuypers[1,3], Diether Lambrechts [10,11], Pierre Marquet[14], Virginie Mathias[15,16], Marion Rabant[7,17], Minnie M. Sarwal[8], Aleksandar Senev [1,18], Tara K. Sigdel[8], Ben Sprangers[1,3], Olivier Thaunat[16,19], Claire Tinel[1,2,20], Thomas Van Brussel[10,11], Amaryllis Van Craenenbroeck [1,3], Elisabet Van Loon [1,3], Thibaut Vaulet[1], Francesca Bosisio [5] & Maarten Naesens [1,3] ✉

Rejection remains the main cause of premature graft loss after kidney transplantation, despite the use of potent immunosuppression. This highlights the need to better understand the composition and the cell-to-cell interactions of the alloreactive inflammatory infiltrate. Here, we performed droplet-based single-cell RNA sequencing of 35,152 transcriptomes from 16 kidney transplant biopsies with varying phenotypes and severities of rejection and without rejection, and identified cell-type specific gene expression signatures for deconvolution of bulk tissue. A specific association was identified between recipient-derived *FCGR3A*+ monocytes, *FCGR3A*+ NK cells and the severity of intragraft inflammation. Activated *FCGR3A*+ monocytes overexpressed *CD47* and *LILR* genes and increased paracrine signaling pathways promoting T cell infiltration. *FCGR3A*+ NK cells overexpressed *FCRL3*, suggesting that antibody-dependent cytotoxicity is a central mechanism of NK-cell mediated graft injury. Multiplexed immunofluorescence using 38 markers on 18 independent biopsy slides confirmed this role of FcγRIII+ NK and FcγRIII+ nonclassical monocytes in antibody-mediated rejection, with specificity to the glomerular area. These results highlight the central involvement of innate immune cells in the pathogenesis of allograft rejection and identify several potential therapeutic targets that might improve allograft longevity.

Despite the development of potent immunosuppressive therapy, rejection remains a prime cause of long-term graft failure after kidney transplantation, representing an unmet need to better understand its underlying mechanisms[1,2]. Rejection of the allograft results from the complex interaction between immune cells, soluble molecules and the kidney cells[3]. Kidney transplant rejection is currently classified[4] into two distinct categories based on the spatial distribution of the infiltrating leukocytes and the presumed underlying allorecognition process: T cell-mediated rejection (TCMR), defined by tubulointerstitial inflammation or arteritis following T lymphocyte activation and

migration to the allograft, and antibody-mediated rejection (ABMR), primarily characterized by microvascular inflammation (MVI) and complement activation induced by the binding of donor-specific antibodies to the donor endothelium[4,5].

This dichotomy in the clinically used Banff classification is challenged by accumulating insights in the heterogeneity of rejection and the alloreactive potential of the innate immune system[3,6]. For example, the frequent observation of histological features of ABMR in absence of donor-specific antibodies (DSA) is puzzling[7] and can potentially be explained by antibody-independent mechanisms[8–11]. Second, therapeutic strategies directed at canonical pathways have shown mixed results. For ABMR, consensus on the ideal treatment is lacking, as several therapies targeting the humoral immune system such as anti-CD20 monoclonal antibodies have not shown long-term benefit[12]. For TCMR, T cell targeted therapies such as CTLA4-Ig, a T cell costimulation blocker[13], improve short-term functional outcomes but lack response in an important subset of patients[14,15], highlighting the need for better preventive and therapeutic treatment options. Third, important heterogeneity in the composition of the inflammatory infiltrate in rejection was observed, independent of the spatial distribution of the infiltrating leukocytes that is used for rejection classification[16]. Altogether, these aspects highlight the need for more precise phenotyping of the immune cell populations involved in kidney transplant rejection subtypes.

Herein, we aimed to uncover the transcriptional, phenotypic and spatial profile of immune cell infiltration in kidney transplant rejection. First, we performed single-cell RNA sequencing of 16 kidney allograft biopsies with varying phenotypes and degrees of rejection (from no rejection to severe rejection). We derived and validated a cell-specific gene expression signature matrix that was used for deconvolution of the cellular heterogeneity in several independent bulk transcriptomic datasets. Finally, we developed a custom multiplex immunofluorescence assay to obtain a high-resolution phenotypical and spatial profile of the inflammatory infiltrate in 18 independent allograft biopsies.

## Results
### Single-cell analysis of kidney allograft rejection
To phenotype the cell subtypes present in transplant rejection, we analyzed 16 allograft biopsies with in total 35,152 cells that passed quality control (Fig. 1a, Supplementary Fig. 1a, Table S1), without major batch effects between samples (Supplementary Fig. 1b) and with an average of 2197 cells per sample (Supplementary Fig. 1c). Using canonical lineage markers[17], all structural compartments of the human kidney were represented in each biopsy (Fig. 1b, c). Epithelial cell populations included *PLA2R1*+ podocytes, *LRP2*+ proximal tubular (PT) cells, *UMOD*+ loop of Henle (LOH) cells and *TMEM213*+ intercalated cells (IC). Vascular cell populations were *TAGLN*+ vascular smooth muscles or pericytes (vSMp), several *PECAM1*+ endothelial cells (EC) populations, including *PLVAP*+ peritubular capillary (ECptc), *CLDN5*+ vasa recta (ECvr), and *EMCN*+ glomerular (ECg) endothelial cells. In addition to these structural cells, several immune cell clusters were identified (Fig. 1b). Lymphoid cell populations encompassed *CCR7*+ naïve CD4 T cells and *CCR7*- memory CD4 T cells, $GZMK_{low}$ effector CD8 T cells, $GZMK_{high}$ effector memory CD8 T cells (CD8effmem) and *MKI67*+ CD8temra cells. Two clusters of natural killer (NK) cells, *FCGR3A*+ and *FCGR3A*- were identified at this resolution as well as a single cluster of *CD19*+ B cells (Fig. 1b, d). Two myeloid cell clusters were detected at this resolution: one cluster with high expression of *CD14* (CD14+ Mono/Macro) and one with high expression of *FCGR3A* encoding for FcγRIIIa (*FCGR3A*+ Myeloid) (Fig. 1b, d). Tubular cells and more particularly PT cells were the most abundant cell types, amounting up to a mean of 54% of total cells in each individual data set, whereas EC represented up to 7% and immune cells represented 30%, in line with previous reports (Fig. 1e)[17,18].

### Leukocyte populations in the kidney allograft are mainly of recipient origin
We next sought to determine whether the identified immune cell populations were of donor or recipient origin by assessing the expression of sex-linked genes as indicator of the origin of cellular populations in transplantations with a gender mismatch between donor and recipient. Five female-to-male transplantations (Supplementary Fig. 3a) and one male-to-female transplantation (Supplementary Fig. 3b) were present in our cohort, of which the cells were assigned as donor- or recipient-derived by the expression of Y chromosome-encoded *ZFY* and *DDX3Y* and female-specific *XIST* and *TSIX*, involved in X chromosome inactivation[19]. The percentage of cells expressing sex-linked genes was heterogeneous among the cell types, which is in line with previous reports[19]. Cell types composing the renal architecture were mostly donor-derived, but a fraction of glomerular endothelial cells (5%) expressed recipient-derived genes, which could reflect endothelial chimerism[20]. Leukocyte populations mostly expressed recipient-derived sex-linked genes, indicating recipient-derived immune cell infiltration, although a small proportion (1.65%) of *CD14*+ Mono/Macro expressed donor-derived genes. This corroborates previous findings on myeloid cells originating from the donor organ, and suggests that myeloid cells may exhibit tissue residency after transplantation[21]. Finally, in our scRNAseq cohort, CD8temra cells were equally recipient- and donor-derived.

### The proportion of *FCGR3A* + NK cells and *FCGR3A*+ myeloid cells is associated with graft inflammation
Next, we investigated which immune cell subtypes are implicated in kidney transplant rejection subtypes. We first stratified the biopsies using a data-driven clustering method producing a phenotypic reclassification of kidney transplant rejection[22]. As depicted on the polar plot generated by this method, biopsies were categorized in four different groups: non-rejection (NR) DSA negative group (NR DSA-, $N = 4$), TCMR ($N = 1$), NR DSA+ ($N = 8$) and ABMR DSA+ ($N = 3$). No biopsy in this cohort was categorized as mixed TCMR-ABMR or DSA-histology of ABMR (ABMRh). We also evaluated the association between the frequencies of the different cell types identified by scRNAseq analysis and the calculated[22] inflammation severity (Fig. 1f). Strikingly, among immune cells, only the frequencies of *FCGR3A*+ myeloid cells and *FCGR3A*+ and *FCGR3A*- NK cells significantly correlated with the severity of inflammation in the kidney allograft (Fig. 1g).

To confirm these findings in a representative cohort of 224 kidney allograft biopsies (GSE147089) with different rejection phenotypes and a wide range of inflammation severity, we derived cell type-specific gene expression signatures from the single-cell clusters using CIBERSORTx. In this aim, we generated a kidney transplant biopsy-derived signature matrix encompassing 18 cell types (KTB18) and estimated the cellular fractions (Fig. 2a)[23]. Pseudobulk samples derived from our single-cell cohort demonstrated good overall correlation between the real cell fractions and the predicted proportions from the CIBERSORTx pipeline (median Pearson $r = 0.83$, Supplementary Fig. 4a). Given that the estimated *FCGR3A*- NK cell fraction did not correlate significantly with the captured fraction, we grouped the NK cell subsets for further deconvolution analyses (Supplementary Fig. 4b). In addition, we demonstrated a good correlation between the deconvoluted cell fractions and the observed cellular densities in multiplex immunofluorescence staining (CD3, CD163, NKp46) on FFPE tissue from 81 allograft biopsies (Supplementary Fig. 4c, d), further corroborating that our deconvolution matrix KTB18 reliably predicts the cellular composition of kidney allograft biopsies based on bulk transcriptomic data.

We next applied the same data-driven clustering method as used for the scRNAseq cohort[22] on the bulk transcriptomics dataset ($N = 224$). This indicated a good distribution of the biopsies on the polar plot, covering the entire spectrum of rejection phenotypes

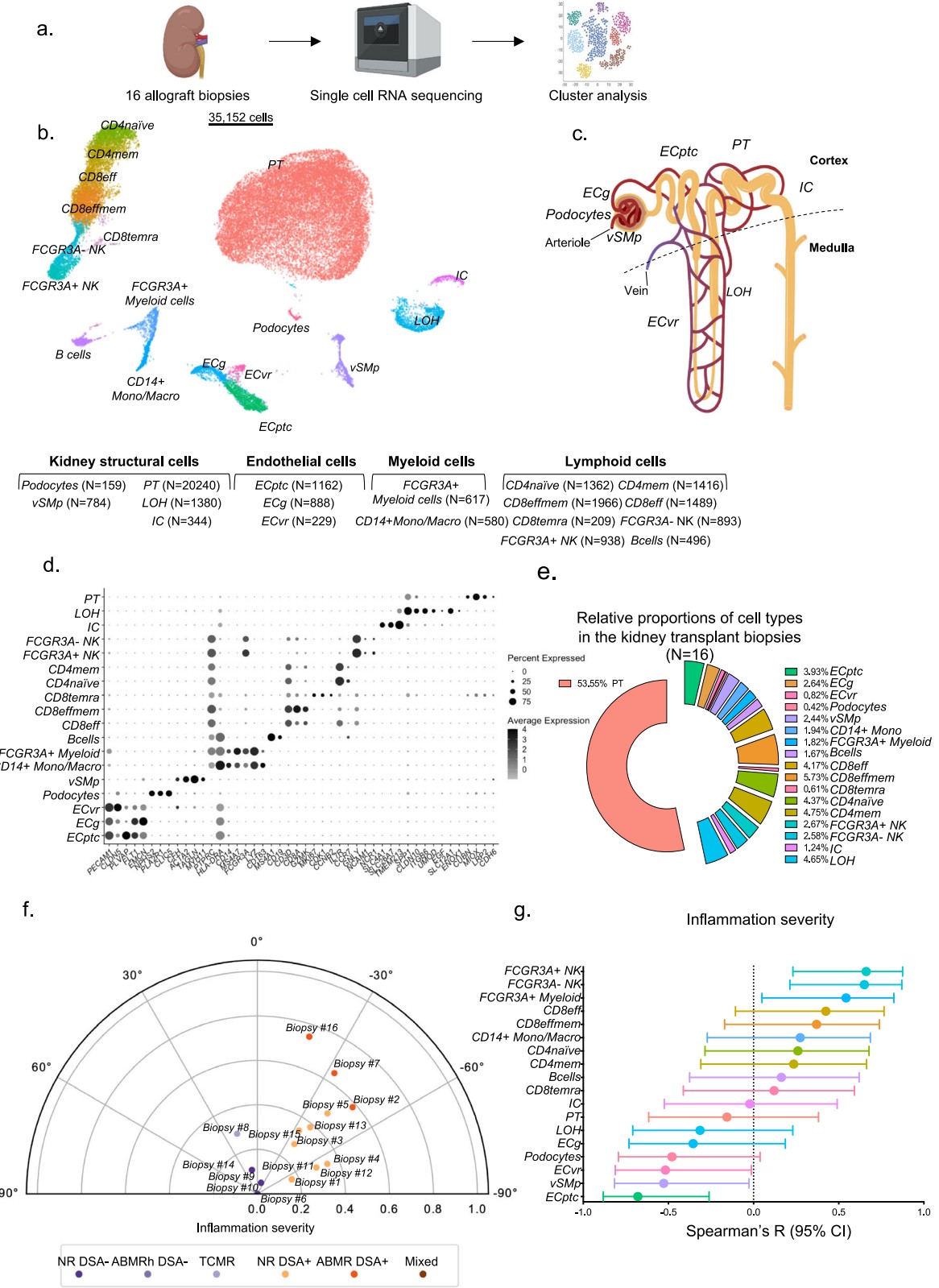

(Fig. 2b). We observed that the cell types best associated with the severity of inflammation were nonclassical monocytes and NK cells, as well as classical monocytes, macrophages and various CD8 + T cell subsets (Fig. 2c). Monocytes, macrophages and CD8 + T cells correlated both with tubulo-interstitial inflammation (tubulitis and interstitial inflammation) and with inflammation in the vascular compartment (glomerulitis, peritubular capillaritis, intimal arteritis) (Fig. 2d). In contrast, NK cell infiltration associated with inflammatory lesions in the vascular compartment but not with tubulitis or interstitial inflammation. In addition, NK cells were increased only in biopsies with ABMRh, whereas nonclassical monocytes were increased in both ABMRh and TCMR (Fig. 2e). Infiltration of classical monocytes and macrophages was increased in all rejection phenotypes but was higher in TCMR compared to ABMRh. Finally, CD8 + T cells were only significantly increased in biopsies with TCMR, either pure or mixed. We confirmed these results in public dataset GSE36059 in which the

**Fig. 1 | Identification of the cell subtypes in kidney allograft biopsies by single-cell RNA seq. a** Experimental Approach. Kidney allograft biopsies were used for scRNAseq ($n = 16$ biopsies from 14 patients). **b** Uniform Manifold Approximation and Projection (UMAP) plot of 35,152 cells passing QC filtering. The main kidney cell types are represented, including loop of Henle (LOH), podocytes, vascular smooth muscle and pericytes (vSMp), proximal tubule (PT), intercalated cells (IC), three endothelial cell subsets comprising vasa recta (ECvr), glomerular (ECg) and peritubular capillaries (ECptc), myeloid cells, and lymphoid cells. **c** Schematic of a juxtamedullary nephron showing relevant cell types and associated vasculature. **d** Dot plot showing average gene expression values of canonical lineage markers

(log scale) and percentage of major cell types represented in the whole dataset and UMAP plot. **e** Relative proportions of the 18 different cell types identified in the kidney transplant biopsies. **f** Polar plot of the 16 biopsies[22]. The biopsies were reclassified into 4 groups: non-rejecting donor-specific antibody (NR DSA)-negative (NR DSA-), NR DSA-positive (NR DSA+), T cell-mediated rejection (TCMR), DSA-positive antibody-mediated rejection (ABMR DSA+). **g** Spearman's correlation coefficient represented by a dot with 95% confident interval represented by the error bars of the correlation between inflammation severity (radius on the polar plot)[22] and the frequency of the different immune cells as a proportion of the total population. Panels a and c were created using biorender.com.

presence of DSA was taken in account for ABMR classification (Fig. 2f), illustrating that DSA status slightly impacts on transcriptional changes in biopsies with ABMRh.

## Phenotypic characterization of the cellular composition of rejection at protein level

To characterize the immune infiltrate in rejection at the protein level, we next performed spatially resolved, single-cell, multiplexed immunofluorescence (MILAN) using a broad panel of immune cell markers on 18 independent kidney transplant biopsies (Fig. 3a, Supplementary Tables 2, 3). For each biopsy, a routine hematoxylin and eosin staining was performed. On a serial slide, the multiplexed immunofluorescent staining was performed before digital reconstruction of the cell populations (Fig. 3b).

Following quality control and clustering using the main phenotypic markers across the included samples, we mapped 555,479 DAPI+ cells to 17 phenotypes including 5 types of renal epithelial cells (PT cells, AQP1+ tubule cells, CD138+ tubule cells, distal tubule cells and AQP1+ PanCK+ tubule cells), 9 immune cell subtypes: 3 myeloid (macrophages, CD1c+ dendritic cells [DC] and S100+ DC), 5 lymphoid (B cells, CD4 regulatory T cells (Tregs), FcγRIII+ NK cells, CD4 effector T cells (Teff) and CD8 Teff) and an MPO+ neutrophil subtype that represented 1.77% of the total cells (Fig. 3c, d). Of note, no cluster corresponding to neutrophils was observed in the scRNAseq dataset. In addition, 26.64% of the total cells were annotated as "not otherwise specified" (NOS) encompassing all the different cell types for which no markers were included in the multiplexing panel (e.g., podocytes, mesangial cells, etc.).

Stratifying the biopsies according to the Banff classification, we noticed that only certain immune types were significantly increased in either TCMR or ABMR rejection phenotypes compared to biopsies without rejection. In particular, macrophages and FcγRIII + NK cells were significantly increased in ABMR, whereas neutrophils, CD1c+ DC and Teff cells were increased in TCMR (Supplementary Fig. 5). Confronting immune cell proportions with inflammation severity (Fig. 3e), only FcγRIII+ NK cells, neutrophils, macrophages and CD8+ Teff proportions correlated with inflammation severity (Spearman correlation, $P = 0.0038$, $P = 0.0097$, $P = 0.0208$ and $P = 0.0344$, respectively), confirming our previous findings at the transcriptomic level (Fig. 3f, g). To further explore the role of myeloid cell activation in transplant rejection, we reintegrated and subclustered all myeloid cell types and distinguished 13 subclusters: CD163+ macrophages, CD68+ macrophages, CD209+ regulatory macrophages, neutrophils, FcγRIII+ CD14-nonclassical monocytes, FcγRIII+ CD14+ intermediate monocytes, FcγRIII- CD14+ classical monocytes, CD141+ cDC1, CD1c+ cDC2, IRF8+ inflammatory DCs, other DCs (characterized only by CD11b) and a not-otherwise-specified (NOS) monocyte cluster (Fig. 4a). Importantly, only the NOS monocytes, MPO+ neutrophils, CD14+ classical monocytes and FcγRIII+ nonclassical monocytes significantly correlated with inflammation severity (Spearman correlation, $P = 0.0002$, $P = 0.0072$, $P = 0.0094$ and $P = 0.0175$ respectively, Fig. 4b, c), highlighting the relevance of these innate immune cells in kidney transplant rejection.

## FcγRIII+ NK cells and FcγRIII+ monocytes relate to vascular inflammation

Having identified an important role for monocytes, FcγRIII+ NK cells, neutrophils and CD8+ Teff, we next investigated the spatial distribution of these cellular populations in kidney transplant biopsies, according to rejection subtype. For this purpose, the proportion of immune cells was measured in the glomerular, vascular, large vascular, tubular and interstitial compartments of the kidney transplant biopsies. In ABMR and mixed rejection, FcγRIII+ NK cells, neutrophils and FcγRIII+ monocytes specifically infiltrated glomeruli and vascular compartments. In contrast, TCMR was hallmarked by infiltration of CD8+ T cells and CD14+ monocytes in the tubulo-interstitium and CD8+ T cells in larger vessels (Fig. 5a).

Glomerulitis is characterized by an increase in the number of mononuclear cells in the glomerular capillary lumina. FcγRIII+ NK cells, FcγRIII+ nonclassical monocytes and neutrophils were significantly enriched inside glomeruli, whereas CD14+ classical monocytes were mainly identified outside the glomeruli (Fig. 5b, c). To investigate differences in the cellular composition of the glomerulus between rejection subtypes, the distance of each cell type from the glomeruli was measured and compared (Fig. 5d). In ABMR and mixed rejection, but not in TCMR, the percentage of FcγRIII+ NK cells, neutrophils and FcγRIII+ monocytes were most present in the glomeruli and rapidly dropped at 100μm away from the glomeruli. These results were concordant with our finding that NK cells and *FCGR3A+* myeloid cells were specifically associated with glomerulitis lesions (Fig. 2d). Moreover, still in ABMR, the proportion of CD14+ monocytes and NOS monocytes increased with distance from the glomeruli. In TCMR, there was less heterogeneity in the spatial distribution of immune cells.

## *FCGR3A+* monocytes specifically express activating Leukocyte Immunoglobulin Like Receptors

Given this confirmation of the involvement of *FCGR3A+* monocytes in solid organ allograft rejection and the need for deeper insight in their activation mechanisms in humans, we then reintegrated all myeloid cells encompassed in our transcriptomic single-cell dataset. We observed 8 clusters: *FCGR3A+ CD14-* nonclassical monocytes, *FCGR3A+ CD14+* intermediate monocytes, *FCGR3A- CD14+* classical monocytes, as well as *CD163+* and *CD68+* macrophages, *THBD+ CLEC9A+* cDC1, *CD1C+ CLEC9A+* cDC2, *MPO+ ITGAX+* neutrophil-like cells (Fig. 6a, b). The proportion of *FCGR3A+ CD14-* nonclassical monocytes significantly correlated with inflammation severity, while other myeloid subtypes did not (Fig. 6c). In pseudotime trajectory analysis, excluding neutrophils and DCs, we observed two distinct trajectories (Fig. 6d): *CD14+* monocytes connected with either *FCGR3A+* monocytes or *CD68+* macrophages cells, which subsequently connected to *CD163+* macrophages. Profiling of marker genes such as microvascular adhesion markers along these trajectories confirmed the functional annotation, as FcγRIII+ nonclassical monocytes are known to patrol along endothelium[24] (Fig. 6e). Besides increasing endothelial adhesion marker expression, *FCGR3A+* monocytes also expressed innate allorecognition markers such as *CD47* and Leukocyte Immunoglobulin Like Receptors (LILR) family genes[25,26] progressively along the

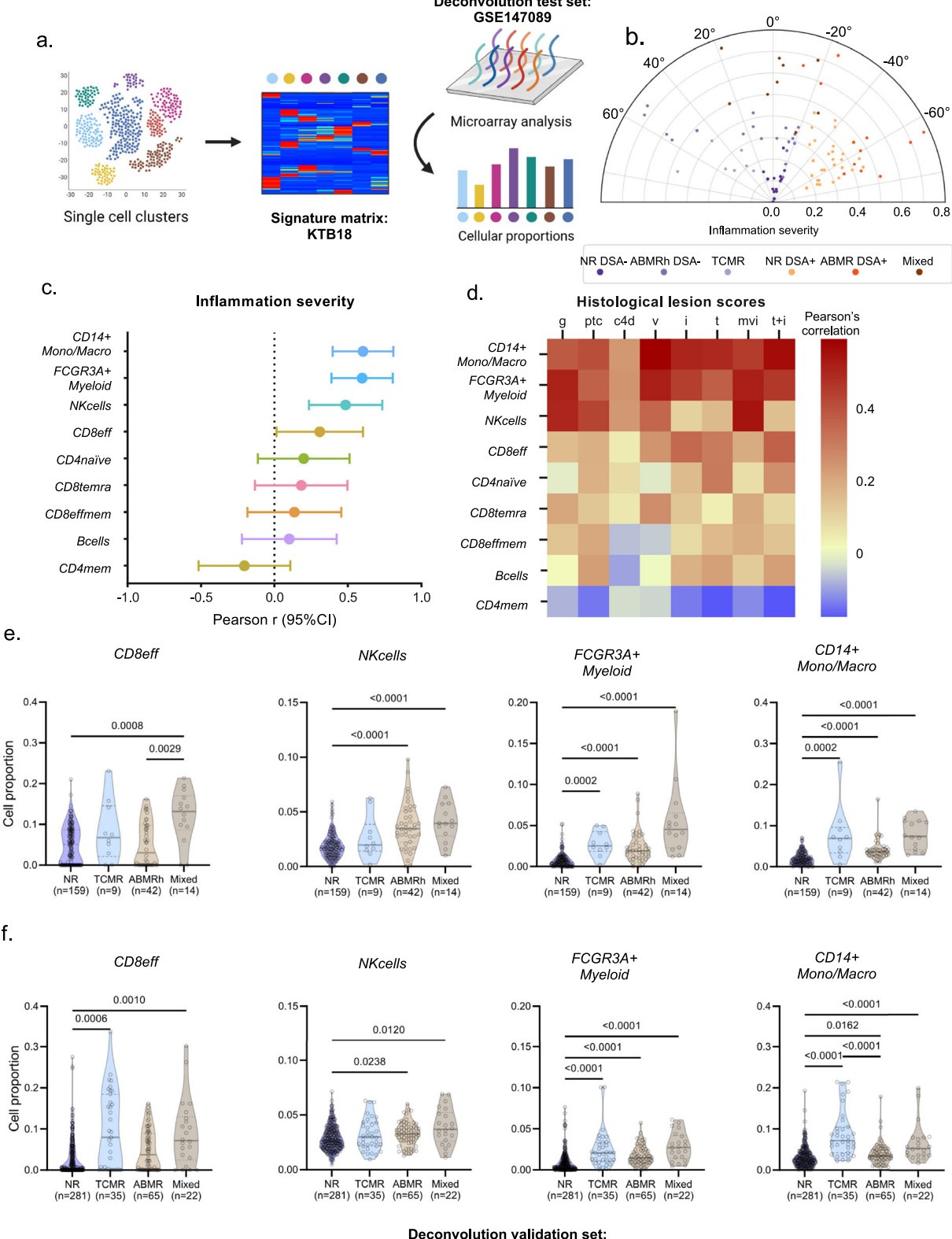

**a.** Single cell clusters → Signature matrix: KTB18 → Deconvolution test set: GSE147089, Microarray analysis → Cellular proportions

**b.** Inflammation severity

NR DSA-  ABMRh DSA-  TCMR  NR DSA+  ABMR DSA+  Mixed

**c.** Inflammation severity — Pearson r (95%CI)

**d.** Histological lesion scores — Pearson's correlation

**e.** CD8eff, NKcells, FCGR3A+ Myeloid, CD14+ Mono/Macro

**f.** CD8eff, NKcells, FCGR3A+ Myeloid, CD14+ Mono/Macro

Deconvolution validation set: GSE36059

trajectory. In contrast, increased expression of complement-related (*C1QA*, *C1QB*, *C1QC*), phagocytosis-related (*LAMP1*, *MSR1*, *OLR1*) and antigen presentation-related (*CD74*, *HLA-DMA*, *HLA-DRA*) genes was observed in *CD68*+ and *CD163*+ macrophages. Overall, this indicates that macrophages mainly assume complement secretion as well as scavenger function and antigen presentation, whereas nonclassical monocytes are equipped with both antigen-dependent cytotoxicity

(FcγRIII) and receptors suggested to be involved in innate allorecognition[25,26].

## Transcriptional changes in *FCGR3A*+ monocytes and *FCGR3A*+ NK cells during allograft rejection

Given that both *FCGR3A*+ monocytes and *FCGR3A*+ NK cells enrichments were associated with inflammation severity, we investigated the

**Fig. 2 | CIBERSORTx deconvolution confirms enrichment of *FCGR3A+* cells in biopsies with rejection. a** Experimental Approach. scRNAseq-derived signature matrix KTB18 was used for deconvolution of the dataset GSE147089 encompassing 224 transcriptomes from kidney biopsies. Created using biorender.com. **b** Polar plot of the 224 biopsies. The biopsies were reclassified into 6 groups: non-rejecting donor-specific antibody (NR DSA)-negative (NR DSA-), NR DSA-positive (NR DSA+), T cell-mediated rejection (TCMR), antibody-mediated rejection DSA-positive (ABMR DSA+), ABMR, DSA-negative histology of ABMR (ABMRh DSA-), and mixed rejection. **c** Pearson's correlation coefficient is represented by a dot, with 95% confident interval shown as error bars, for the correlation between

inflammation severity and frequency of different immune cells as a proportion of the total cell population ($n = 224$ biopsies). **d** Correlogram of the indicated immune cell proportions and Banff histological lesions using Pearson's correlation. Colors indicate correlation coefficient. i, interstitial inflammation; ptc, peritubular capillaritis; v, vasculitis; mvi (g+ptc), microvascular inflammation; t, tubulitis. **e, f** Frequency of the indicated cell subsets measured by deconvolution and stratified according to clinical outcome. The difference between groups was assessed by a two-tailed Kruskal-Wallis test and multiple comparisons using the Dunn's test. Data were obtained from the GSE147089 test set[23] **f** Data derived from the public GSE36059 validation set[83].

transcriptional changes occurring in these two cell types upon ABMR. In *FCGR3A+* monocytes, a robust transcriptional response of 11 differentially expressed genes (DEG) was observed in NR DSA+ (Fig. 7a) that coincided with that of the ABMR group. Immediately apparent was the strong upregulation of the chemokine *CXCL10* together with upregulation of the two IFN-γ-inducible genes *GBP5, WARS* and two other transcripts, *TNFSF10* and *SLC31A2* (logFC>1.5 in ABMR). Conversely, we observed a downregulation of the chemokines *CCL3* and *CCL17* but also *CD163, JUN*, and *FN1* (logFC < −1.5 in ABMR group) suggesting a shift in cytokine signaling by *FCGR3A+* monocytes. The expression of these genes related to inflammation severity (Fig. 7c). We confirmed these findings in independent scRNAseq datasets, reintegrating two ABMR biopsies[21] as well as two mixed ABMR-TCMR biopsies[27] with four biopsies from patients showing no rejection[21] (Supplementary Fig. 6a). In *FCGR3A+* nonclassical monocytes, *CXCL10, GBP5, WARS, SLC31A2* and *TNFSF10* were overexpressed in the ABMR and mixed rejection cases, while *JUN* and *CCL3* were decreased (Supplementary Fig. 6b, c). Expression of *CCL17* and *CD163* was observed in only one sample and *FN1* not expressed at all in this public dataset.

Analysis of *FCGR3A+* NK cells displayed much less DEG in comparison between NR DSA- and ABMR samples. Only *FCRL3* showed a logFC>1.5 in ABMR compared to NR DSA- (Fig. 7b). *FCRL3*, a gene increased after FcγRIII engagement during ABMR[28], correlated positively with inflammation severity (Fig. 7d). In the validation set, overexpression of *FCRL3* in ABMR was observed in the two ABMR samples but was less clear in mixed rejection (Supplementary Fig. 6d, e).

## Characterization of *FCGR3A+* monocytes and *FCGR3A+* NK cell-cell communication

We next used CellChat to quantitatively infer intercellular communication networks[29] and noticed more than 60 significantly enriched pathways (Fig. 8a) contributing to cell-to-cell communications. A limited number of communications were attributed to *FCGR3A+* NK cells (Fig. 8b), mainly represented by the secretion of the macrophage migration inhibitory factor MIF and the expression of its receptors, CD74 + CXCR4 or CD74 + CD44 in the other immune cells. This is in line with literature stipulating that *FCGR3A+* NK cells are maturated NK cells that progressively decreased their cytokine secretion but that are more cytotoxic compared to their *FCGR3A-* counterparts[30].

In contrast, *FCGR3A+* myeloid cells were predicted to communicate with almost all other cell types within the kidney except for PT cells and IC. A large number of communications in *FCGR3A+* myeloid cells was observed in which GALECTIN represented an additional broad signaling pathway together with TNF and CXCL signaling (Fig. 8c). Specifically, myeloid cells exerted a privileged communication with *FCGR3A+* NK through LGALS9-HAVCR2 signaling (Fig. 8d). *HAVCR2* encodes for Tim-3 protein, a marker of NK cell activation that enhances IFN-γ secretion in response to galectin-9 within the allograft[31,32]. In line with this, *CXCL10*, an IFN-γ-inducible gene, is particularly upregulated in *FCGR3A+* monocytes during rejection and mainly interacts with T cells through *CXCR3*, but also with ECptc through *ACKR1* (DARC, also called the Duffy antigen/chemokine receptor) (Fig. 8e, f).

## *FCGR3A+* NK and *FCGR3A+* monocytes strongly interact in ABMR

Finally, we assessed the cell-cell contact signaling, as predicted in the transcriptomic datasets, in the MILAN spatial protein expression dataset. At the transcriptional level, the most active pathways were represented by APP-CD74, CD99-CD99 and CD8-class I HLA interactions followed by ITGB2-ICAM2 and LILRB1-class I HLA (Fig. 9a). Focusing on *FCGR3A+* NK cell-cell contacts, we noticed that NK cells mainly interacted with CD8 Teffmem through class I HLA-CD8 and with *FCGR3A+* myeloid cells through class I HLA-LILRB1 but also ITGB2-ICAM and CD99-PILRA (Fig. 9b, left panel). *FCGR3A+* myeloid cells mainly responded to autocrine signaling through more than 30 pathways including LILRB1-S100A but also LILRB1-class I HLA and CD4-class II HLA, PECAM1-PECAM1 and ITGB2-ICAM pathways. They also displayed two unique pathways to communicate with *FCGR3A* + NK cells: HLA-E-KLRC1 and HLA-E -CD94:NKG2A (Fig. 9b, right panel). Since CellChat analysis identified *FCGR3A+* myeloid cells as the dominant communication "hub" within the allograft, mainly communicating with CD8 T cells and *FCGR3A+* NK cells, we finally tested whether FcγRIII+ myeloid cells would differentially induce recruitment of CD8 T cells and FcγRIII+ NK cells according to the rejection subtype. For this, we performed a neighborhood analysis at the proteomic level to characterize which immune cells were specifically located around FcγRIII+ monocytes within the allograft (Supplementary Fig. 2). We noticed a strong and significant enrichment of FcγRIII+ NK cells in ABMR or mixed rejection biopsies, but not in TCMR or NR biopsies. In contrast, FcγRIII+ monocytes were mainly surrounded by CD8 T cells in TCMR (Fig. 9c). Altogether, these results suggest that FcγRIII+ monocytes might distinguish non-self through LILRs expression and secrete TNF, galectin-9 and CXCL10. This activation and cytokine secretion could support the alloimmune responses together with FcγRIII+ NK cells in antibody-mediated context, or with CD8 T cells in antibody-independent context. To better characterize the role of monocytes in supporting CD8+ T cells alloreactivity, we set up an in vitro model in which CD8+ T cells, purified from an HLA-A2 negative healthy volunteer, were cocultured with a human cell line expressing only the allogeneic molecule HLA-A2 in presence or absence of monocytes from the same healthy volunteer. The proliferating HLA-A2 specific CD8+ T cells were detected using CellTrace dye dilution (Supplemental Material and Methods, Fig. S8a, b). After 4 days of coculture with monocytes, the HLA-A2 specific CD8+ T cells showed a trend towards an increase of proliferation (Fig. S8c), as well as stronger cytotoxic profiles as illustrated by higher level of expression of CD107a, Granzyme-B and IFN-γ expression (Fig. S8d) compared to CD8 + T cells cultured without monocytes. In line with these phenotypical data, HLA-A2 specific CD8+ T cells cocultured with monocytes showed an increased capacity to kill HLA-A2-expressing renal epithelial cells as compared to non-specific CD8+ T cells or HLA-A2 specific CD8+ T cells cultured in absence of monocytes. Altogether, these results further support our hypothesis that monocytes may support the expansion and acquisition of a cytotoxic profile by allo-specific CD8 + T cells.

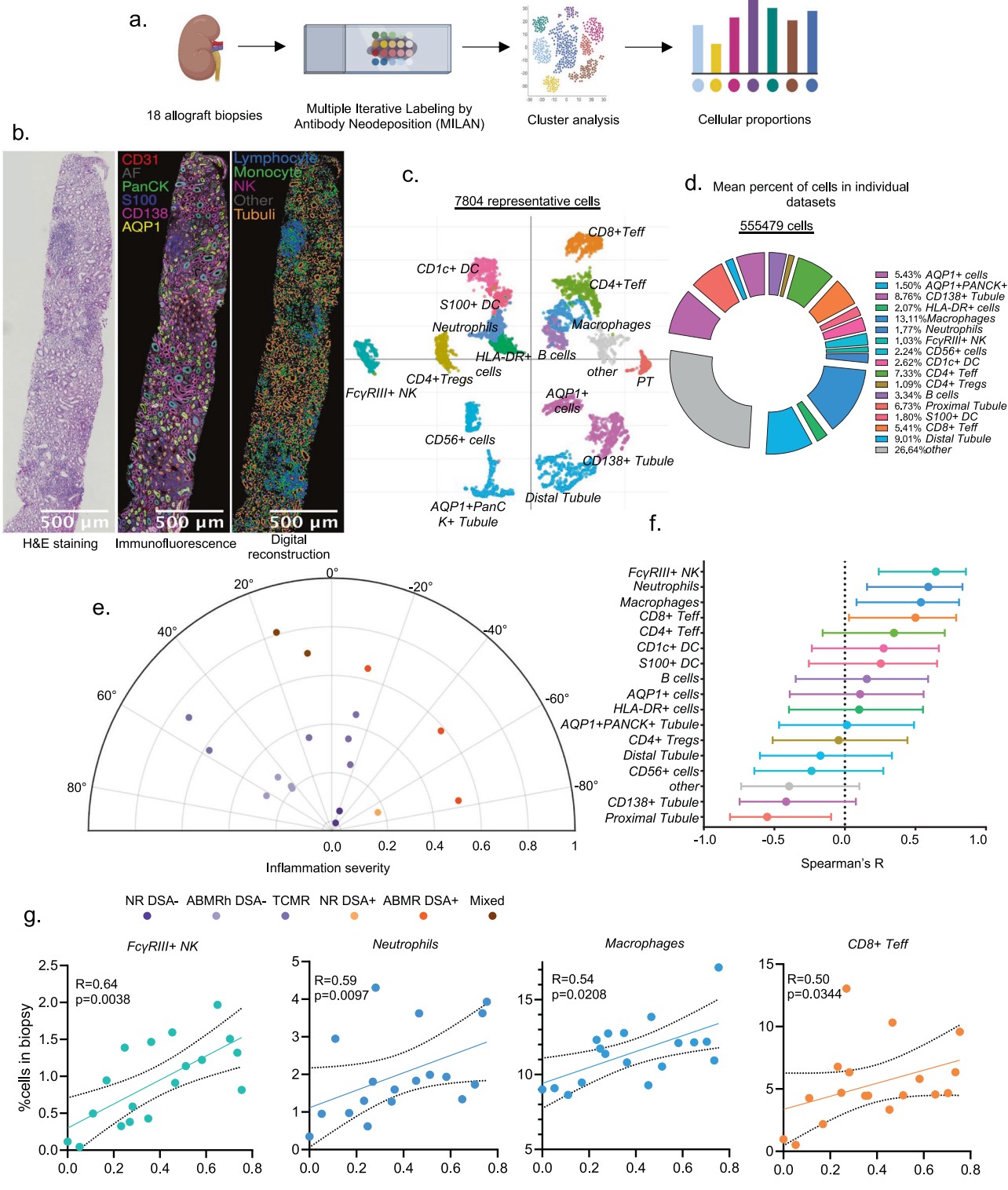

**Fig. 3 | The cellular composition of rejection characterized at the protein level using Multiple Iterative Labeling by Antibody Neodeposition (MILAN).**
**a** Experimental Approach. MILAN was used for characterization of infiltrating cells in kidney biopsies at the protein level in 18 kidney biopsies. Created using Biorender.com. **b** Representative sample illustrating the MILAN analysis with hematoxylin and eosin (H&E) staining (left), composite fluorescence image of 5 markers (plus autofluorescence, AF) after image processing (center), digital reconstruction of the sample highlighting the main cell populations of interest. **c** Uniform Manifold Approximation and Projection (UMAP) plot of 7804 representative cells. **d** Relative proportions of the 555,479 cells identified in the kidney transplant biopsies (*n* = 18).

**e** Polar plot of the 18 biopsies. The biopsies were reclassified into 6 groups: non-rejecting donor-specific antibody (DSA) negative (NR DSA-) and DSA positive (NR DSA+), T-cell mediated rejection (TCMR) and antibody-mediated rejection DSA positive (ABMR DSA+) and DSA negative (ABMRh DSA-) and Mixed rejection. **f** Spearman's correlation coefficient represented by a dot with 95% confident interval represented by the error bars of the correlation between inflammation severity and the frequency of different immune cells as a proportion of the total cell population (*n* = 18 biopsies). **g** Correlations between the frequency of indicated cell types and inflammation severity. Spearman's correlation coefficient and two-tailed p-value are indicated.

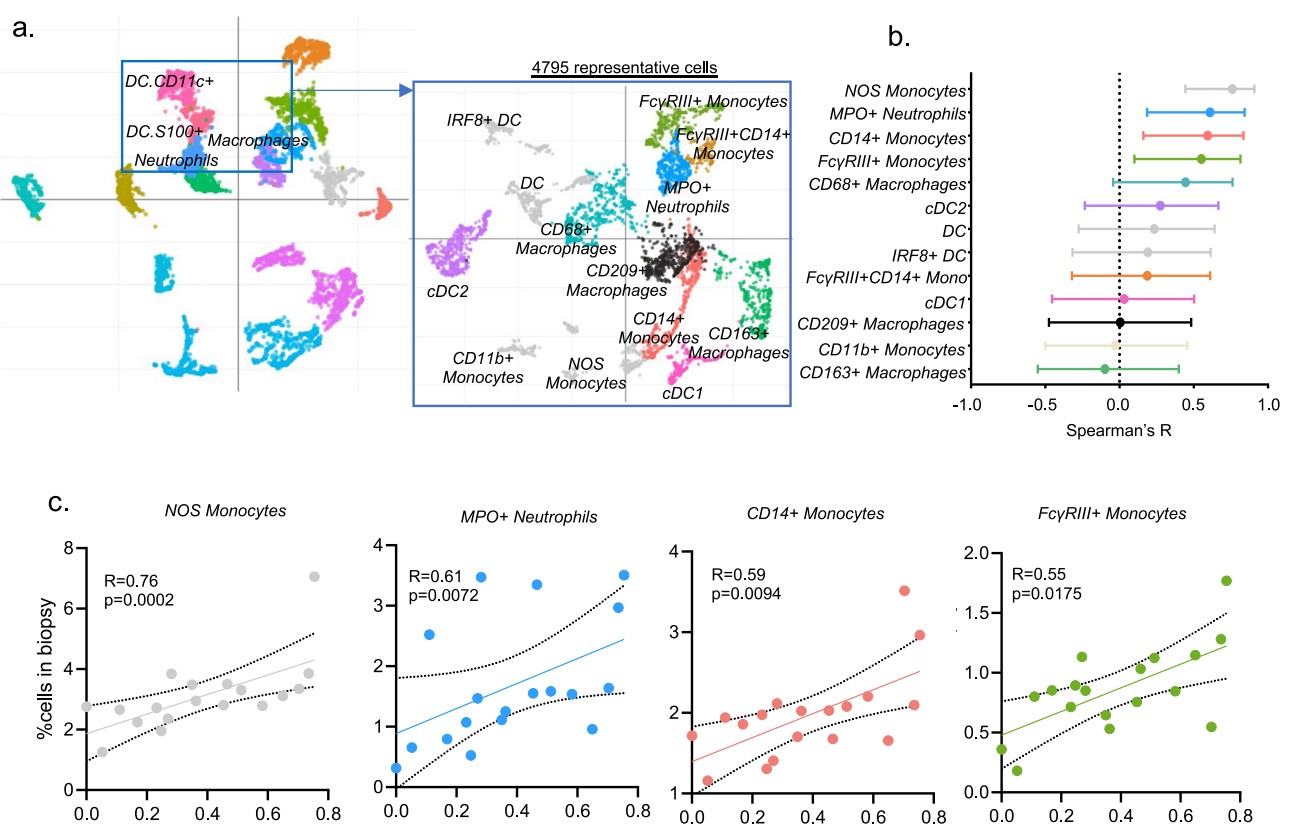

**Fig. 4 | Distinct monocytes populations are enriched in inflamed biopsies.**
**a** Reintegration of DC, neutrophils and macrophages populations and Uniform Manifold Approximation and Projection (UMAP) plot of the 4795 corresponding cells. **b** Spearman's correlation coefficient represented by a dot with 95% confident interval represented by the error bars of the correlation between inflammation severity and the frequency of different immune cells as a proportion of the total cell population (*n* = 18 biopsies). **c** Correlations between the frequency of indicated cell types and inflammation severity. Spearman's correlation coefficient and two-tailed p-value are indicated.

## Interactions between *FCGR3A*+ NK and *FCGR3A*+ myeloid cells and structural kidney cells

To evaluate cell communications between the *FCGR3A*+ immune cells and the structural kidney cells, we next reintegrated PT cells, endothelial cells, podocytes and immune cells of interest: *FCGR3A*+ NK and *FCGR3A*+ myeloid cells (Fig. 10a). Four different clusters were distinguished within PT cells, corresponding to the 3 segments already described from kidney cortical area to medulla (PT_S1, PT_S2 and PT-S3 respectively, Fig. 10a–c). Interestingly, a cluster corresponding to injured tubular cells expressing *HAVCR1* and *VCAM1* (Fig. 10b) was detected[33]. Regarding endothelial cells, in addition to the previously described clusters (ECptc, ECg and ECvr), we detected a cluster expressing high levels of *VCAM1*, suggesting activation as previously reported[21]. We performed CellChat analysis assessing both secreted signaling and cell-cell contact using *FCGR3A* + NK and *FCGR3A*+ monocytes as source cells. Very limited cellular communications were detected in PT cells clusters (Fig. 10d), which is in line with Fig. 10e confirming that no significant interaction was detected between the tubules and *FCGR3A*+ immune cells. In contrast, secreted signaling unraveled that *FCGR3A*+ monocytes mainly interact with NK cells through *LGALS9* coding for galectin-9 inferred secretion (Fig. 10e). We previously showed that anti-HLA DSA triggers chemokine and cytokine production in kidney transplant recipients' serum, independent of histological lesions and we demonstrated that this secretion could be induced by FcγRIII+ cells in an in vitro model of ADCC mimicking anti-HLA DSA binding on endothelium[34]. Here, using the same in vitro model, we found that non-classical monocytes significantly secreted more galectin-9 after anti-HLA DSA recognition (Fig. S8a–d).

Therefore, we measured the expression of Tim-3, the galectin 9-receptor in FcγRIII+ NK cells in kidney biopsies using multiplexed immunofluorescence (MILAN). We observed that Tim-3 expression is increased in FcγRIII+ NK cells closely located near FcγRIII+ monocytes in acute rejection context (Fig. S8e, f) suggesting that galectin-9 secreted by FcγRIII+ monocytes could trigger NK cell activation in rejection.

In addition, *VCAM1*+ EC (EC_Injured) overexpress *ACKR1* and *TNFRSF1A*, making them potential targets of monocyte-derived CXCL10 and TNF but also of NK-derived CCL5. Given that the recognition of an Fc fragment by the FcγRIII receptor induces Syk dependent pathways triggering cytokine secretion, we investigated whether a Syk inhibitor could block the secreted signaling using the in vitro model of ADCC. The addition of R406, the active form of fostamatinib, to the NK/non-classical monocytes/GENC coculture completely abrogated the secretion of TNF, CCL5 and CXCL10 in the presence of HLA DSA, suggesting that Syk inhibitors could alleviate the cytokine release of NK cells and monocytes in ABMR (Fig. S9a, b).

Regarding cell-cell contacts between *FCGR3A*+ NK and *FCGR3A*+ monocytes, the main interactions were driven by *ITGB2* and *ITGA4* expression. The basal expression of their ligands *ICAM1* and *ICAM2* in ECvr, Ecptc and Ecg strengthened by *VCAM1* overexpression in EC_Injured could explain this preferential cell-cell contact (Fig. 10e). We did not identify any injury cluster of podocytes given their limited number.

## Discussion

In this study, by performing scRNASeq on 16 human kidney transplant biopsies, we identified the major infiltrating immune cells

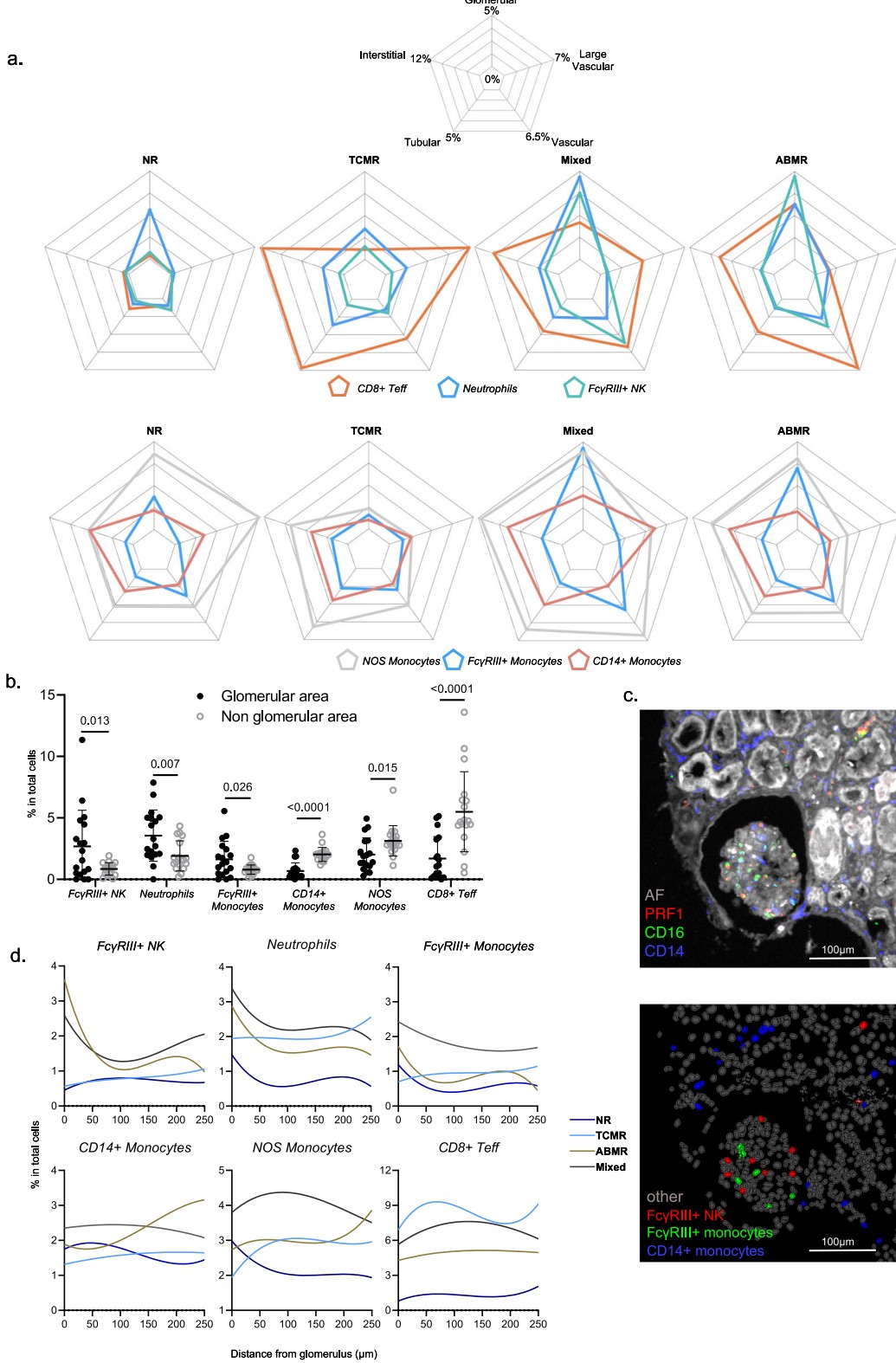

**Fig. 5 | Immune cells are differently compartimentalized in the kidney allograft. a** Radar plots showing the percentage of indicated cell populations as a proportion of the total cell population measured in the glomerular, tubular, vascular, large vascular and interstitial compartments. **b** Percentage of indicated cells measured in the glomerular vs. non-glomerular area. The mean ± SEM is depicted.The difference between groups was assessed by the two-tailed *p* value calculated with Mann-Withney test. **c** Representative case showing a composite fluorescent image of 3 markers (plus autofluorescence, AF) after image processing (top), and the resulting digital reconstruction highlighting the cell types of interest (bottom). The reader should note that FcγRIII+ NK cells and FcγRIII+ nonclassical monocytes infiltrate the glomerulus whereas classical monocytes are located outside of it. **d** Enrichment of 6 different immune cell types based on their distance to the closest glomerulus. The y-axis represents the percentage of cells (from the total cell population) belonging to a given cell type at a maximum given distance (x-axis) from the closest glomerulus. Different samples from the same Banff classification were pulled to get a single line per category.

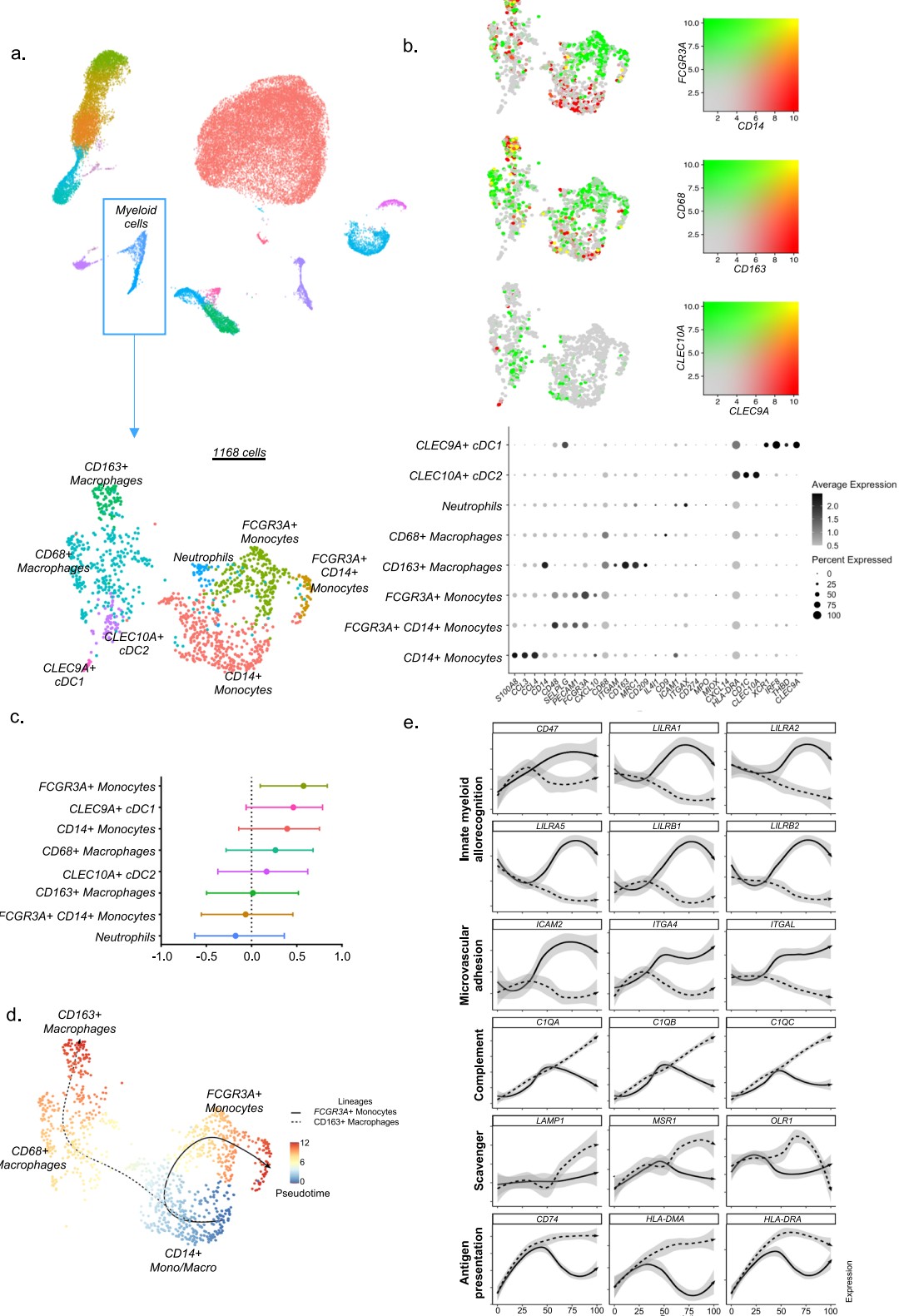

**Fig. 6 | *FCGR3A*+ monocytes are involved in innate myeloid allorecognition.** **a** All myeloid cells identified in the 16 scRNAseq samples were reintegrated in a UMAP plot of 1,168 myeloid cells. **b** Dot plot demonstrating average gene expression values of canonical lineage markers (log scale) and percentage of myeloid cell types represented in the UMAP plot. **c** Correlations between inflammation severity (the radius on the polar plot[22]) and the proportion of each myeloid subset relative to the total number of cells assessed in each biopsy. Spearman's correlation coefficient represented by a dot with 95% confident interval represented by the error bars. **d** Pseudotime trajectories for monocytes and macrophages based on Slingshot, showing the common branch of *CD14*+ monocytes differentiating into either *FCGR3A*+ monocytes or *CD68*+, then into *CD163*+ macrophages. The light grey error band represents the SEM of the gene expression **e** Profiling of marker genes along these trajectories.

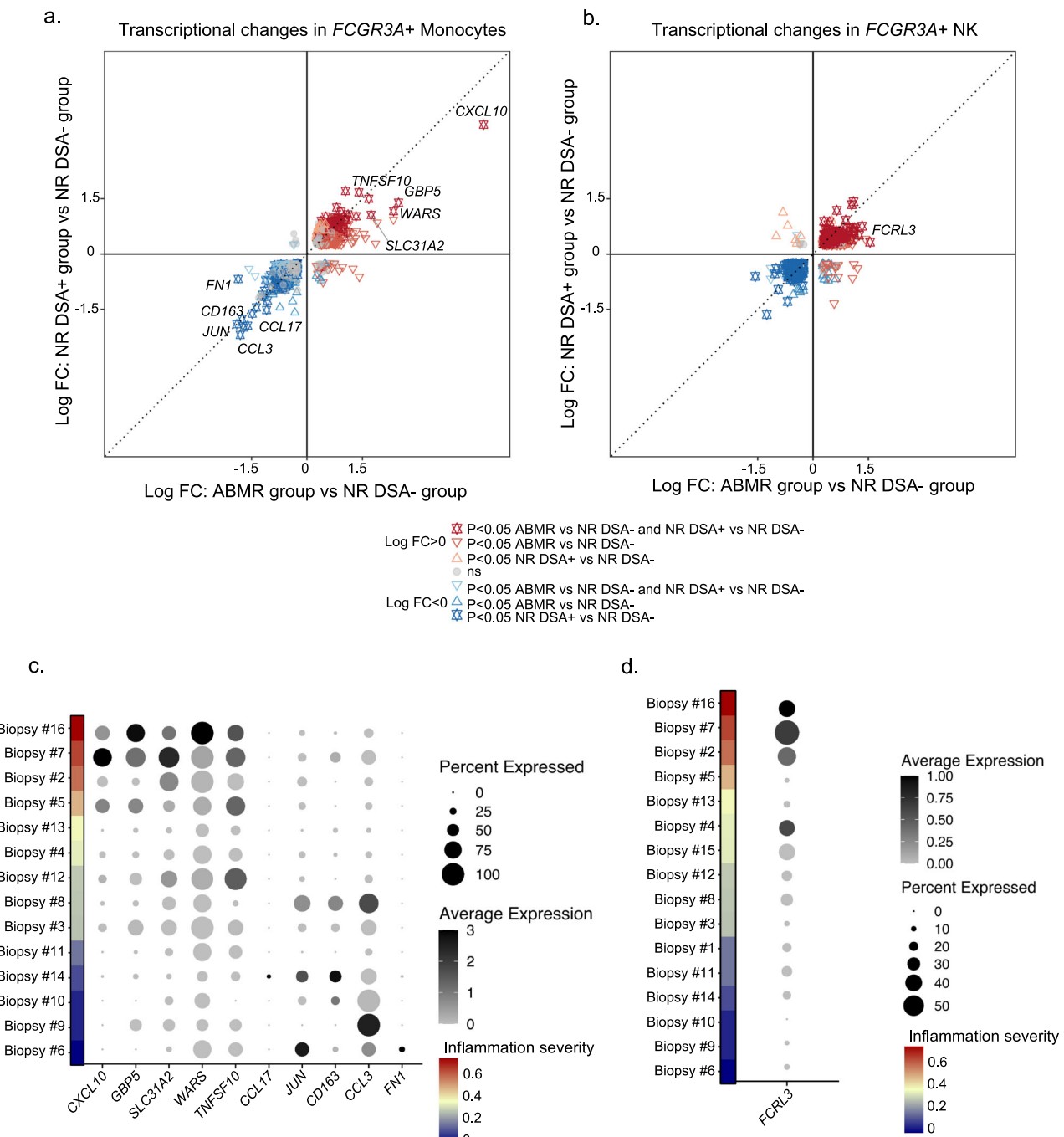

**Fig. 7 | Transcriptional changes in *FCGR3A*+ NK and *FCGR3A*+ monocytes during rejection. a**, **b** The differentially expressed genes (DEGs) of *FCGR3A*+ cells were identified by the FindMarker function in Seurat using Wilcoxon test, according to the diagnostic phenotype of the samples (*n* = 16). **a** The fold change-fold change (FC-FC) plot compares the transcriptional differences within *FCGR3A*+ monocytes, between non-rejecting donor-specific antibody (NR DSA-) and NR DSA+ (y-axis) and antibody-mediated rejection (ABMR) (x-axis). The highlighted DEGs represent transcripts with log FC > 1.5 or log FC < −1.5 in ABMR cases compared to NR DSA-.

**b** FC-FC plot comparing the DEGs in *FCGR3A*+ NK cells between NR DSA- and NR DSA+ (y axis) or ABMR (x axis). The highlighted DEGs represent transcripts with log FC > 1.5 or log FC < −1.5 in ABMR. **c, d** Dot plots demonstrating average gene expression values of selected genes (log scale), and percentage of *FCGR3A*+ Monocytes expressing indicated genes (**c**) or *FCGR3A*+ NK cells expressing indicated genes (**d**), according to inflammation severity (the radius on the polar plot[22]) represented by the rank of the biopsies in the plot.

relevant in allograft rejection. To our knowledge, this is the largest single-cell cohort of human kidney transplant biopsies to date. The cell fractions that were identified, and their respective prevalence, were consistent with previous single-cell studies[17,21,35,36] and were statistically correlated with detailed clinicopathological phenotypes and inflammation severity. Gene expression signatures and a cell-specific deconvolution matrix KTB18 that were derived from the

single-cell populations allowed to estimate the cellular fractions in a transcriptomic cohort of 224 independent allograft biopsies. This confirmed the correlation between FcγRIII+ NK cell and FcγRIII+ monocyte infiltration and inflammation severity. We demonstrated that FcγRIII+ NK cells and FcγRIII+ monocytes are increased in ABMR and microvascular inflammation, while TCMR and tubulo-interstitial inflammation associated with CD8+ T cells, monocytes, and

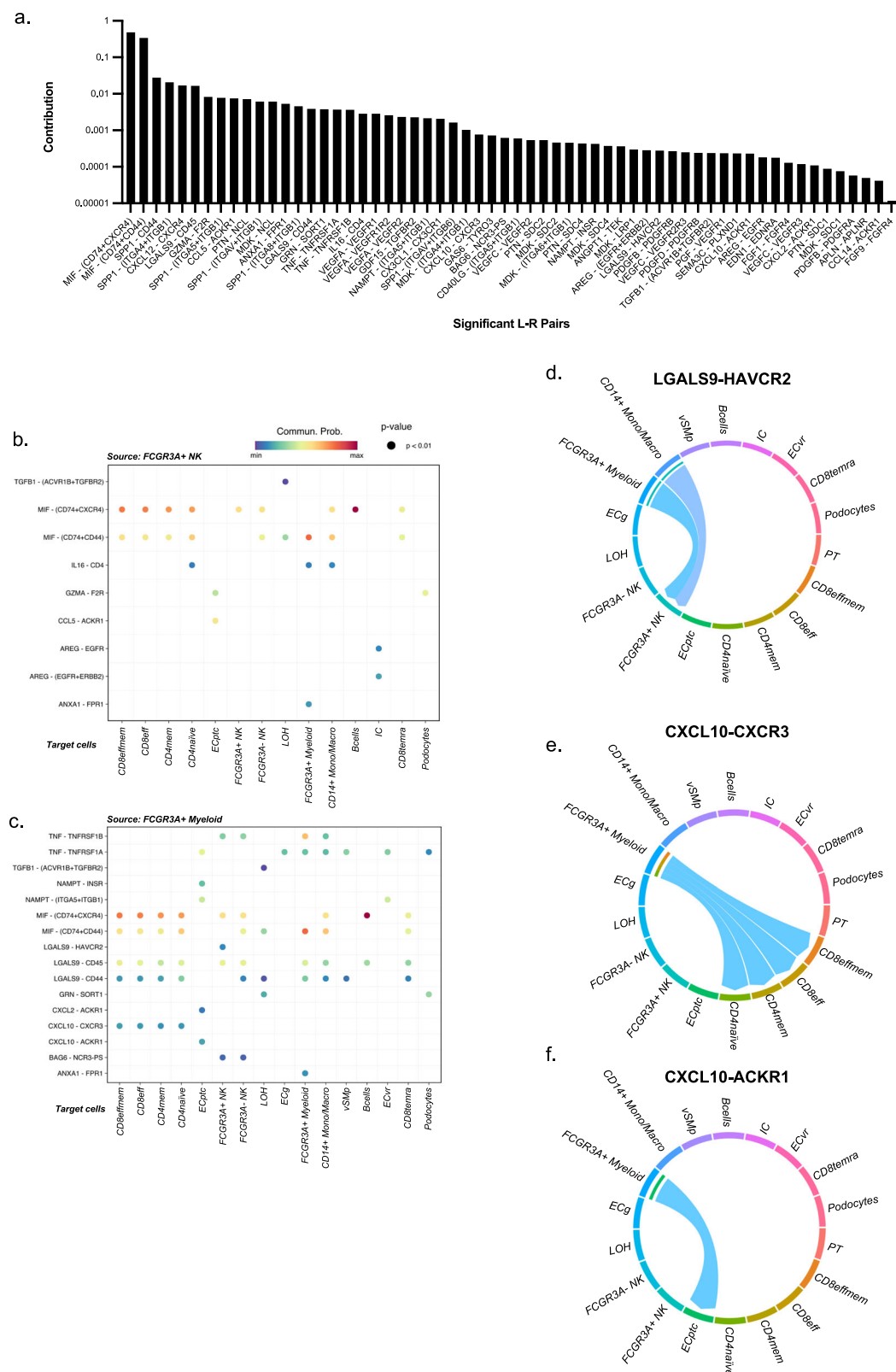

**Fig. 8 | *FCGR3A*+ monocytes present specific communication patterns. a** All the significant communications contributing to secreted signaling inferred by CellChat analysis are depicted. **b** Ligand-receptor pairs that significantly contribute to the secreted signaling from FcγRIII+ NK to all other cell types. **c** Ligand-receptor pairs that significantly contribute to the secreted signaling from *FCGR3A*+ monocytes to all other cell types. **d** The inferred LGALS9-HAVCR2 signaling network. **e, f** The inferred CXCL10 signaling network encompassing CXCL10-CXCR3 (**e**) and CXCL10-ACKR1 ligand-receptor pairs (**f**).

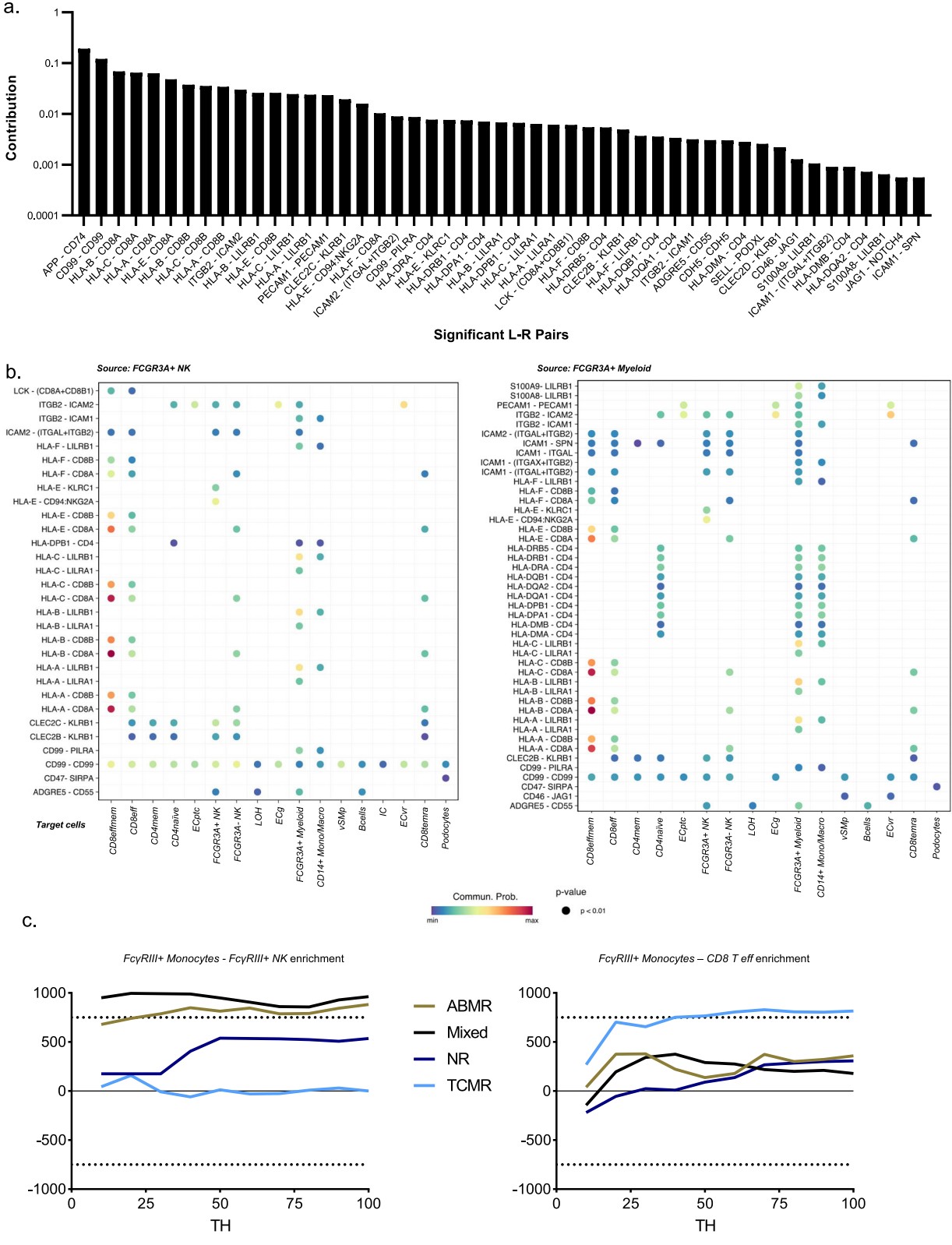

**Fig. 9 | *FCGR3A*+ monocytes neighborhood analysis. a** All the significant communications contributing to cell-to-cell signaling inferred by CellChat analysis are depicted. **b** Ligand-receptor pairs that significantly contribute to the cell-to-cell signaling from *FCGR3A*+ NK (left panel) or *FCGR3A*+ monocytes (right panel) to all other cell types. **c** Neighborhood analysis results showing the enrichment of cell-

cell interactions between *FCGR3A*+ monocytes and *FCGR3A*+ NK cells or CD8 T eff for the different Banff classification groups. The y-axis represents the number of random cases with a higher (if above 0) or smaller (if below 0) number of interactions than the observed data (see methods). The x-axis represents the distance at which one cell is considered to be in the neighborhood of another cell.

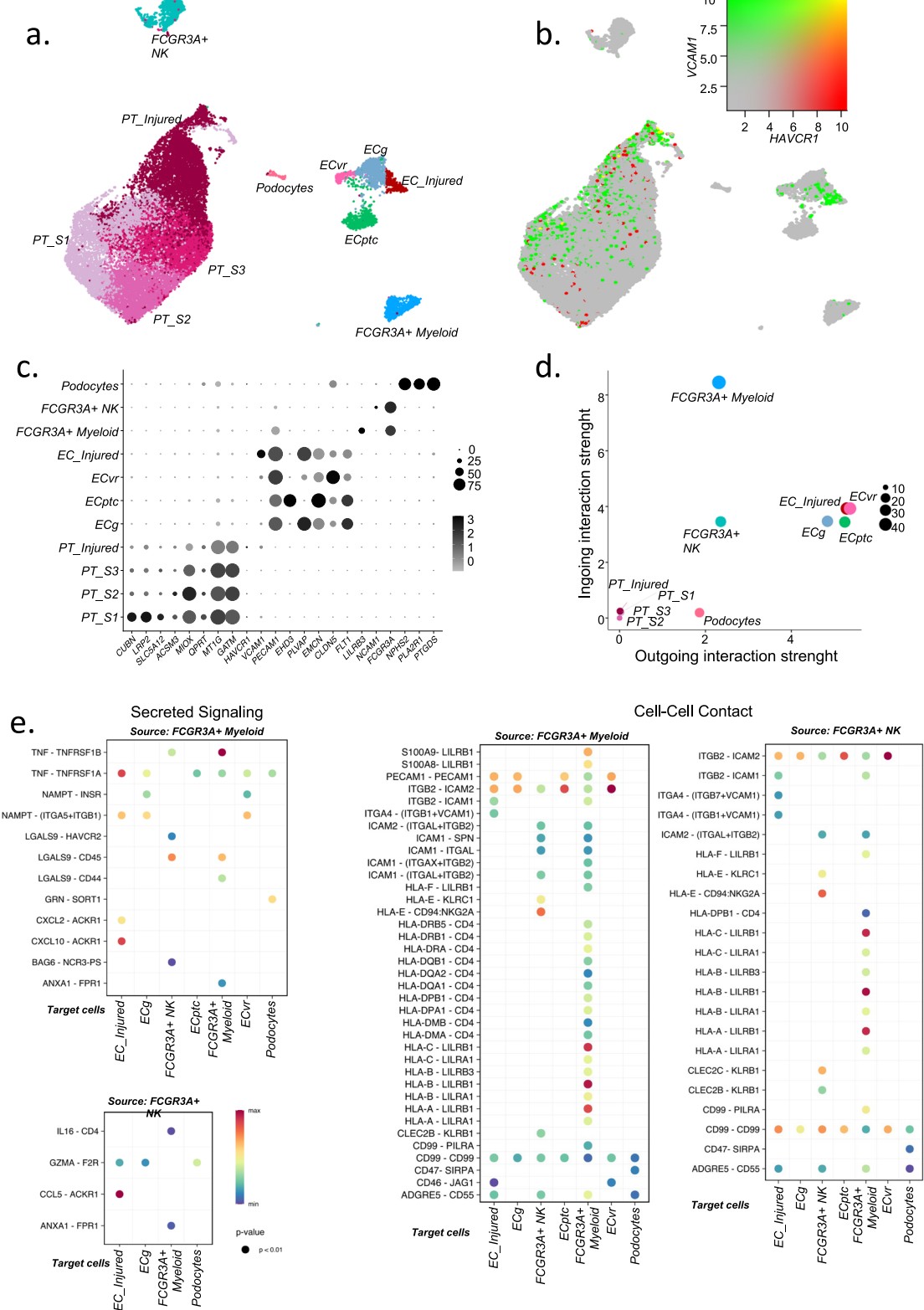

**Fig. 10 | FCGR3A+ cells mainly interact with the endothelium by direct contact but also through secreted mediators. a** *FCGR3A*+ cells were reintegrated with endothelial/epithelial cells in a UMAP plot. **b** UMAP depicting the concomitant expressions of two markers of injury: *VCAM1* and *HAVCR1*. **c** Dot plot demonstrating select average gene expression values (log scale) and percentage cell types represented in the UMAP plot. **d** CellChat analysis was performed and the number of incoming and outgoing ligand-receptor interactions is plotted per cell type. **e** All the significant communications contributing to cell-to-cell signaling inferred by CellChat analysis are depicted: ligand-receptor pairs that significantly contribute to the secreted effectors (left panels), or cell-to-cell direct contact from *FCGR3A*+ NK or *FCGR3A*+ monocytes to all other cell types (right panels).

macrophages. Using multiplex immunofluorescence analysis, we confirmed the differential involvement and spatial distribution of the innate immune cell infiltrate in the vascular vs. tubulo-interstitial rejection subtypes. Finally, these studies indicate molecular and spatial cell-cell interactions between the FcγRIII+ monocytes and FcγRIII+ NK cells in microvascular rejection. Our results collectively point to the central involvement of recipient-derived innate immune cells, in particular NK cells and monocytes, in kidney transplant rejection, which is usually assumed to be primarily mediated by adaptive immunity.

In our study, FcγRIII+ NK cells were correlated with inflammation severity and more particularly with microvascular inflammation. Among NK cells, FcγRIII+ NK cells represent a cytotoxic subset, whereas FcγRIII- NK cells harness more regulatory functions[37]. FcγRIII, encoded by *FCGR3A*, is a surface receptor that interacts with the constant fraction of target-bound antibodies. Notably, Sablik et al. reported an increase of *FCGR3A* expression on circulating NK cells in chronic active ABMR[38]. This activation is hallmarked by transcriptional changes and notably the overexpression of *FCRL3*[28], the release of cytolytic granules and apoptosis-inducing ligands that lead to the destruction of the antibody-bound cellular target[39] in a process termed antibody-dependent cellular cytotoxicity (ADCC). Moreover, NK cells can also sense the absence of HLA-I molecules through the KIR receptors and thus recognize missing self, which can contribute to and even trigger microvascular inflammation[9,40,41]. In our study, transcriptional changes of these cells were very limited in ABMR patients, which suggests that these cells primarily interact with the allograft through *FCRL3* induction and thus ADCC. Further studies are required to investigate the cellular interactions and interplay of potential other activating stimuli that lead to FcγRIII+ NK cell activation in microvascular inflammation.

Next to demonstrating the role of FcγRIII+ NK cells in kidney transplant rejection, our study also identified the main subsets of monocytes, macrophages, and dendritic cells in kidney allografts: CD14+ classical monocytes, CD14+ FcγRIII+ intermediate monocytes, CD14- FcγRIII+ nonclassical monocytes, CD68+ macrophages and CD163+ macrophages but also CLEC9A+ cDC1 and CLEC10A+ cDC2. Whereas the main function of classical monocytes is phagocytosis and scavenging, allowing the elimination of microorganisms and/or abnormal cells, FcγRIII+ nonclassical monocytes present less phagocytosis activity but more ADCC capability[42]. FcγRIII+ monocytes infiltration within the allograft was positively correlated with inflammation severity, suggesting that nonclassical monocytes can play a pathogenic role during rejection, most likely by ADCC after anti-donor antibody fixation, and by amplifying CD8+ T cells allogeneic responses. Interestingly, FcγRIII+ monocytes represented the main myeloid cells expressing the newly described receptors involved in innate allorecognition, such as *CD47*. Recipient-derived CD47+ infiltrating myeloid cells could sense allogeneic SIRPα expressed by donor cells, as was recently suggested[25]. Moreover, after CD47 priming, murine monocytes overexpressed paired immunoglobulin-like receptors (PIRs) that directly bind class I MHC molecules[43] and can thus provide a second hit in case of MHC mismatch[44]. Human orthologs to murine PIRs are leukocyte immunoglobulin-like receptors (LILRs). Among the LILR family, only LILRA1, LILRA2, LILRA3 are known to interact with HLA molecules to elicit an activator signal[45]. In our study, *LILRA1, LILRA2* and *LILRA5* were mainly expressed by FcγRIII+ monocytes, which is further indication that these cells are not only capable of ADCC, but are also able to directly recognize non-self donor cells.

The present study was not designed to address the mechanistic causes of rejection, but the expression patterns of the immune cell infiltrate corroborate recent hypotheses. Recipient-derived antibodies bound to the allograft tissue can enhance immune responses in FcγRIII+ cells. Here, we showed that these same cells are also equipped with either KIRs (for NK cells)[46] or LILRs (for monocytes)[47], which thus could participate in allorecognition not solely restricted to humoral responses as driver of kidney rejection mediated by these FcγRIII+ cells. Indeed, as often in immunological processes, a single effector cannot induce a complete phenotype by itself and the immunological response is far more complex than expected. Similar to what was observed for NK cells, where missing self and ADCC mechanisms can act synergistically[9], the potential synergistic effects of FcγRIII engagement and LILRs activation in FcγRIII+ monocytes need to be addressed. As activator LILRA family downstream signaling relies on ITAM phosphorylation, driving Syk and PI3K pathways[47], this signaling converges on FcγR signaling and mTOR activation. We can thus speculate that pharmacological inhibition of the Syk or mTOR pathway could lead to proportional decrease in both FcγRIII+ monocytes and FcγRIII+ NK cell activation[48,49]. Using an in vitro model of ADCC, Shin et al. showed that steroids and calcineurin inhibitors were effective at reducing IFN-γ production by NK cells compared to mTOR inhibitors[50] but their impact on FcγRIII+ monocytes remains to be addressed. Also, daratumumab, targeting CD38 and depleting NK cells and monocytes, could be considered for treating ABMR, as suggested by Doberer et al. [51]. In addition, a recent report indicated that murine inflammatory monocytes could be targeted using immune-modifying nanoparticles to prevent acute kidney allograft rejection[52]. Building on our human data on the Syk pathway in rejection, we now demonstrate that fostamatinib inhibits FcγR-triggered, Syk-dependent activation[53] and, as such, represents an additional avenue for targeting this previously unexplored mechanism of kidney rejection pathogenesis.

Finally, our findings suggest that recipient-derived nonclassical monocyte infiltration into the graft, followed by FcγRIII engagement and CXCL10 secretion, contributes significantly to the immunological attack against the allograft. FcγRIII+ nonclassical monocytes not only exhibit local cytotoxicity, but also actively contribute to the ongoing pro-inflammatory microenvironment in the allograft. FcγRIII+ monocytes overexpress galecti- 9, a 40-kDa S-type β-galactoside binding lectin which is a known ligand for Tim-3 encoded by *HAVCR2* gene and induced at the NK cell surface after activation. In response to galectin-9, Tim-3+ NK cells enhance IFN-γ production[31]. This finding echoes our recent report showing that coculturing FcγRIII+ monocytes with FcγRIII+ NK cells and glomerular endothelial cells enhanced IFN-γ secretion in vitro[34]. FcγRIII+ monocytes also overexpress the chemokine *CXCL10* together with upregulation of two other IFN-γ-inducible genes, *GBP5* and *WARS,* upon rejection. These three transcripts were previously reported as primary DEGs in bulk transcriptomic analyses of kidney allograft rejection[54] and ABMR[23]. Using microdissection on formalin-fixed kidney allograft biopsies combined with mass spectrometry-based proteomics, it was recently reported that 77 proteins were deregulated in glomerulitis compared to stable grafts, particularly involved in cellular stress mediated by interferons type I and II, leukocyte activation and microcirculation remodeling[55]. Also here, WARS1 protein was overexpressed in leucocytes infiltrating the glomeruli as well as in endothelial cells during ABMR[55,56]. Furthermore, our cell-to-cell communications analysis suggested that *CXCL10* overexpression by FcγRIII+ monocytes may recruit *CXCR3* + T cells and activate *ACKR1*+ endothelial cells composing the peritubular capillaries. This aligns with our previous finding that the FcγRIII engagement of purified nonclassical monocytes co-cultured with primary endothelial glomerular cells induces CXCL10 secretion[57]. Finally, increased expression of CXCL10 is not only observed in the biopsies at time of kidney allograft rejection but also in urine, serving as suitable biomarker of rejection[58–60], and in peripheral blood[57].

Our study has some limitations. For example, in the scRNAseq dataset, the limited number of samples and case mix could have impacted on the results, and less common phenotypes or disease processes may have been missed. The sample size of our biopsy

cohorts, the way biopsies are obtained (primarily cortical tissue), the choice of technique for single cell dissociation and wet lab preparation, and the downstream clustering affects the cellular mix and transcriptional profiles of the cells both in scRNASeq[61] and in MILAN, and explain why in the scRNASeq dataset no distal convoluted tubule or principal cell and only few podocytes were obtained. In addition, we used a non-overlapping set of samples in scRNAseq and MILAN. This also explains some discrepancies between the scRNAseq results and the MILAN multiplex immunofluorescence data. Moreover, in the MILAN-derived dataset, we largely focused on immune cell subtypes, excluding common epithelial cell markers especially from the medulla area (e.g. SLC12A1 for LOH) and injury markers. As a result, epithelial-immune neighborhoods were not fully assessed at epithelial subtypes level. Nevertheless, the main analyses showed robust corroboration of our conclusions by the different techniques applied to different samples sets. Finally, the relation between the histological/molecular phenotype and cellular mix in the biopsies, and the underlying causal pathways of allorecognition or even autoreactivity (DSA, non-HLA antibodies, missing self, etc.) was not assessed and needs further study.

In conclusion, we have used single-cell transcriptomics, deconvolution of bulk transcriptomics by a novel cell-specific signature matrix KTB18, and multiplex immunofluorescence to characterize the immune landscape of kidney transplant rejection. Current paradigms of kidney transplant rejection focus on adaptive immunity, antibody-mediated injury, and T-cell activation, which are the primary targets of immunosuppressive regimens to prevent and treat rejection. We have demonstrated the importance of FcγRIII+ monocytes and FcγRIII+ NK cells, and innate immune crosstalk during rejection. We believe that this work is a major resource for the understanding of the role of these cells and the innate immune system involvement in kidney transplant rejection, and shows promise for developing therapeutic strategies that target these pathways.

## Methods

### Patient population and data collection

Single-cell RNA sequencing (scRNAseq) was performed on a cohort of 16 biopsies from 14 renal transplant recipients followed in the University Hospitals Leuven, Belgium (Supplementary Table 1). All transplantations were performed with negative complement-dependent cytotoxicity crossmatches on T and B cells. Most recipients had a planned indication biopsy with a high clinical probability for humoral rejection. This study included both male and female participants. Sex annotation was based on self-report. For two patients, a follow-up biopsy was included in the study. Additionally, one recipient of an identical twin transplantation was included with a protocol biopsy at 3 months post-transplantation.

The selection of the bulk transcriptomic samples was reported before[23]. Briefly, we included 224 renal allograft biopsy samples from four European transplant centers between June 2011 and March 2017 (University Hospitals Leuven, Belgium; Medizinische Hochschule Hannover, Germany; Centre Hospitalier Universitaire Limoges, France, and Hôpital Necker Paris, France), in the context of the BIOMArkers of Renal Graft INjuries (BIOMARGIN) study (ClinicalTrials.gov number NCT02832661), and the Reclassification using OmiCs integration in KidnEy Transplantation (ROCKET) study. Institutional review boards and national regulatory agencies (when required) approved the study protocol at each clinical center. Each patient contributed one biopsy and gave written informed consent.

For the multiplex immunofluorescence (MILAN) analysis, an independent set of 18 biopsies was included from renal transplant recipients followed in the University Hospitals Leuven, Belgium (Supplementary Table 2). All patients provided written informed consent. This study was approved by the Ethics Committee of the University Hospitals Leuven (S64904).

### Clinicopathological diagnosis of acute rejection subtypes

Two kidney biopsy cores were obtained using a 14-gauge needle under sonographic guidance. One biopsy core was fixed in formalin and embedded in paraffin for standard histopathological assessment. Half of the second biopsy core was used for frozen sections and/or electron microscopy; the remaining half core was used for either scRNAseq or bulk transcriptomic analysis. All biopsies were scored according to the internationally standardized Banff lesion scores[62]. The follow-up of anti-HLA antibodies was systematically monitored in one histocompatibility laboratory (HILA –Belgian Red Cross Flanders); details on this assessment were previously published[7]. Anti-human leukocyte antigen (HLA) donor-specific antibodies (DSA) were assessed retrospectively, taking into account both donor and recipients high-resolution HLA genotyping results. A possible presence of DSA was suspected at a background-corrected median fluorescence intensity value around 500. For the final assignment of DSA, the total sera reactivity of the patients was analyzed. A diagnostic label was awarded to each biopsy based on the presence and severity of these histological lesions and on the DSA status, in concordance with the Banff 2019 classification[4]. In addition, all biopsies were categorized into one of six recently described clinicopathological clusters[22], based on the presence of acute histological lesions and recipient DSA status (available at https://rejectionclass.eu.pythonanywhere.com/). This data-driven classification allows for visualization of the biopsies on a two-dimensional polar plot in which the theta angle associates with the spatial localization of the inflammation, i.e., microvascular inflammation versus tubulointerstitial inflammation. The radius indicates the global severity of the inflammatory infiltrate given by the sum of re-weighted acute lesions scores, scaled to the unit interval (from 0 to 1) and is designated as "inflammation severity"[22].

### Single-cell isolation and single-cell droplet-based RNA sequencing

Biopsies were minced into small pieces with a scalpel and incubated at 37 °C for 5 minutes in freshly prepared dissociation buffer containing 2 mg/mL Collagenase P (Roche) and 0.2 mg/ml DNase I (Roche). Dissociated tissue was harvested and filtered through a 40 μm cell strainer (Flowmi Tipstrainers, VWR) into ice-cold PBS. Cells were collected by centrifugation at 300 g for 5 minutes at 4 °C before resuspension in Red Blood Cell lysis buffer (Merck) for 5 minutes, then centrifuged at 200 g for 5 minutes at 4 °C before resuspension in PBS containing 0.04% UltraPure BSA (AM2616, ThermoFisher Scientific), and finally strained through a 40-μm cell strainer to further remove cell clumps and large fragments. Cell number and viability was measured for the biopsy using Luna Cell counter as previously published[63]. Libraries for scRNAseq were generated using the Chromium Single Cell 5′ library and Gel Bead & Multiplex Kit from 10x Genomics. We aimed to profile 5000 cells per library if sufficient cells were retained during dissociation. All libraries were sequenced on Illumina NextSeq until sufficient saturation was reached. After quality control, raw sequencing reads were aligned to the human reference genome GRCh38 and processed to a matrix representing the UMIs per cell barcode per gene using CellRanger (10x Genomics, v3.1).

### Single-cell RNA-sequencing data analysis

Filtered gene expression matrices generated per sample were merged and analyzed using the Seurat V4 package[64]. Cell matrices were filtered with the following parameters: cells having <400 and >10000 genes detected and having greater than 25% mitochondrial transcripts were excluded (Supplementary Fig. 1a). After filtering, all objects were integrated using 3000 features. Full data set UMAP was generated using Seurat's DimPlot function using the top 17 principal components. Clusters were built using the FindNeighbors and FindClusters functions in Seurat (resolution = 2). Cluster identification was performed using the FeaturePlot function by evaluating the expression of

specific markers in each cluster. Dot plots, violin plots and heatmaps were generated using the DotPlot and VlnPlot functions respectively in Seurat, with normalized counts in the RNA assay as input data. Enriched genes identification was performed using the FindMarkers function in Seurat. A p-value cut-off (p < 0.05) was applied to select significantly enriched genes in indicated cell clusters. Expression levels of significant enriched genes were calculated using AverageExpression function in Seurat according to each patient group as previously defined[22]. Validation scRNAseq dataset was compiled and analyzed as previously described[34,65].

## CellChat and trajectory inference analysis

CellChat uses a mass action-based model for quantifying the communication probability between a given ligand and its cognate receptor and takes into consideration the proportion of cells in each group across all sequenced cells and expressed co-factors (https://github.com/sqjin/CellChat)[29]. A Seurat object encompassing all the cells was used to generate a corresponding CellChat object. The recently reported LILRA and LILRB ligand-receptors pairs[47] were added to the CellChat database. The aggregated cell-cell communication network was calculated by counting the number of links or summarizing the communication probability. The aggregated cell-cell communication network showing the total interaction strength (weights) between all clusters was represented using circle plots. The contribution of each ligand-receptor pair to the overall signaling pathway was computed and the function extractEnrichedLR was used to extract all the significant interactions (L-R pairs) and related signaling genes for all available signaling pathways.

The R package Slingshot was used to explore pseudotime trajectories/potential lineages in myeloid cells[66]. The analyses were performed with DC excluded due to their unique developmental origin. For each analysis, PCA-based dimension reduction was performed with differentially expressed genes of each phenotype, followed by two-dimensional visualization with UMAP. Next, this UMAP matrix was fed into SlingShot with classical monocytes as a root state for calculation of lineages and pseudotime.

## Bulk transcriptomics and deconvolution analysis

Biopsy samples for bulk transcriptomics were immediately stored in Allprotect Tissue Reagent® (Qiagen, Benelux BV, Venlo, The Netherlands). Sample processing was described previously[23]. Briefly, fragmented cRNA was hybridized to the Affymetrix GeneChip Human Genome U133 Plus 2.0 Arrays (Affymetrix), which comprised of 54,675 probe sets covering the whole genome. The resulting image files (.dat files) were generated using the GeneChip® Command Console® Software (AGCC), and intensity values for each probe cell (.cel file) were calculated. The transcriptomics data were handled in accordance with the MIAME (Minimum Information About a Microarray Experiment) guidelines. The data were analysed using TAC software (version 4.0, Thermo Fisher Scientific, Carlsbad, CA, United States) and Bioconductor tools in R (v3.5.3, www.rstudio.com)[67]. The robust multichip average method was performed on the raw expression data (.cel files) to obtain a log2 expression value for each probe set, and batch effect correction was performed for timing of the analysis by use of the LIMMA package[68,69].

We generated a kidney transplant biopsy-derived signature matrix that contains 3110 genes distinguishing 18 kidney and immune cell phenotypes (KTB18) for implementation in the CIBERSORTx deconvolution algorithm[70]. Briefly, the anti-logged gene expression profiles of 10,000 labeled cells from the single-cell dataset, pertaining to 18 distinct cell types, were used as input. Considering that a droplet-based technique was used for single-cell analysis, the *Min. Expression* value was reduced to 0.1 in order to increase reliability of the signature matrix, as recommended by the developers[71]. The number of barcode genes per cell type was held between 300 and 500, resulting in a matrix consisting of 3110 genes. CIBERSORTx was used to deconvolute the predicted cell fractions from the antilogged gene expression data in the bulk transcriptomics cohort. For genes that were represented by several probesets in the microarray platform, the probeset with the highest average expression was used. S-mode batch correction was applied to correct for technical variation between platforms.

## Multiple Iterative Labeling by Antibody Neodeposition (MILAN) and image acquisition

Multiplex immunofluorescent staining was performed according to the previously published MILAN protocol[72]. The antibody panel for MILAN was designed to allow a phenotypic identification of the most abundant immune cell types based on the results from the scRNAseq. An overview of the panel with the 38 markers included and the specifications about the primary and secondary antibodies can be found in Supplementary Table 3. Immunofluorescence images were scanned using the Axio scan.Z1 slidescanner (Zeiss, Germany) at 10X objective with resolution of 0.65 micron/pixel. The hematoxylin and eosin slides were digitized using the Axio scan.Z1 slidescanner in brightfield modus using a 20X objective with resolution of 0.22 micron/pixel. All samples were stained simultaneously. Image-acquisition order was distributed spatially and independently of patient replicates. The stains were visually evaluated for quality by digital image experts and experienced pathologists (FB, YVH, double blinded). Multiple approaches were taken to ensure the quality of the single-cell data. On the image level, the cross-cycle image registration and tissue integrity were reviewed; regions that were poorly registered or contained severely deformed tissues and artifacts were identified, and cells inside those regions were excluded. Glomerular regions and large vessels were manually annotated by an experienced pathologist (FB) on the autofluorescent images. Antibodies that gave low confidence staining patterns by visual evaluation were excluded from the analyses. Image analysis was performed following a custom pipeline. Briefly, flat field correction was performed using a custom implementation of the algorithm previously described[73]. Then, adjacent tiles were stitched by minimizing the Frobenius distance of the overlapping regions. Next, images from consecutive rounds were aligned (registered) following the algorithm previously described[74]. For registration, the first round was used always as fixed image whereas all consecutive rounds were sequentially used as moving images. Transformation matrices were calculated using the DAPI channel and then applied to the rest of the channels. After registration, the performance of the overlapping was evaluated by visual inspection. Samples with tissue folds showed significant misalignments and were manually segmented in different regions. Each region was independently re-registered. Downstream analysis was independently performed for each annotated region. Next, tissue autofluorescence was subtracted using a baseline image with only secondary antibody. Finally, cell segmentation was applied to the DAPI channel using STARDIST[75]. For every cell, topological features (X/Y coordinates), morphological features (nuclear size), and molecular features (Mean Fluorescence Intensity (MFI)) of each measured marker) were extracted.

## MILAN phenotypic identification

MFI values were normalized within each region to Z-scores as recommended in Caicedo et al.[76]. Z-scores were trimmed in the [0, 5] range to avoid a strong influence of any possible outliers in downstream analyses. Single cells were mapped to known cell phenotypes using three different clustering methods: PhenoGraph[77], FlowSom[78], and KMeans as implemented in the Rphenograph, FlowSOM, and stats R packages. While FlowSom and KMeans require the number of clusters as input, PhenoGraph can be executed by defining exclusively the number of nearest neighbors to calculate the Jaccard coefficient which was set to 30. The number of clusters identified by PhenoGraph was then passed as an argument for FlowSom and KMeans. Clustering was performed

exclusively in a subset of the identified cells (50,000) selected by stratified proportional random sampling and using only the 20 markers defined as phenotypic (Supplementary Table 3). The stratification was performed by selecting a number of cells in each sample equal to the relative proportion of the number of cells in that sample in the entire dataset. That is:

$$S_i = S \cdot \frac{N_i}{M}, \text{ where } M = \sum_{i=1}^{P} (N_i)$$

where Si is the number of cells to be sampled for the i-th sample, S is the total number of cells to be sampled (here 50,000), Ni is the number of cells in the i-th sample, and M is the total number of cells in the dataset (sum of all samples, P).

For each clustering method, clusters were mapped to known cell phenotypes following manual annotation from domain experts (FB, YVH, double blinded). If two or more clustering methods agreed on the assigned phenotype, the cell was annotated as such. If all three clustering methods disagreed on the assigned phenotype, the cell was annotated as "not otherwise specified, NOS". Annotated cells were used to construct a template in order to extrapolate the cell labels to the rest of the cells included in the dataset. To that end, a UMAP was built by sampling 500 cells for each identified cell type in the consensus clustering. The complete dataset was projected into the umap using the base predict R function. For each cell, the label of the closest 100 neighbors was evaluated in the UMAP space and the label of the most frequent cell type was assigned. To gain more resolution into monocyte subtypes, cells identified as "dendritic cells" (DC), "macrophages", or "neutrophils" were grouped together, and reclustered again following the same method as described above yet using a set of 13 monocyte-specific markers (Supplementary Table 3, 'monocyte profiling').

### MILAN in-silico microdissection and neighborhood analysis

Next, to gain insights into tissue structure and the interactions between structural cells (tubuli, blood vessels, etc.) and the immune infiltrate, tissue samples were segmented into 5 different regions: glomerular, tubular, vascular (distinguishing large vessels and small vessels), and interstitial. Glomerular regions and large vessels were manually annotated by an experienced pathologist (FB). Small vessels were identified by applying a mask to the CD31 marker. Tubular and interstitial areas were distinguished by training a pixel classifier in QuPath[79].

For neighborhood analysis, a quantitative analysis of cell-cell interactions was performed using an adaptation of the algorithm described in Schapiro et al.[80]. A detailed description of the adapted implementation was previously described[81]. Briefly, for every cell, all the other cells that are located at a maximum distance d were counted. Then, the tissue is randomized preserving the cytometry of the tissue as well as the X and Y coordinates of each cell but permutating the cell identities. This is repeated N times (here N = 1000) which allows to assign an empirical p-value by comparing the number of counts observed in the real tissue versus the number of counts in the randomized cases. Here, we performed the described analysis for different values of the distance d (from 10 to 100 micrometers with a step of 10 micrometers) to show the consistency of the reported results. Since here the main interest was to evaluate interaction partners between immune cells, structural cells (tubular and endothelial cells) were not included in the randomization process to avoid any potential bias introduced by the structure of each sample (Supplementary Fig. 2).

### Multiplex immunofluorescence staining and image processing (Opal)

A multiplex immunofluorescence staining method on paraffin-embedded tissue was recently described using Opal reagents (PerkinElmer, Waltham, MA)[16]. Briefly, tissue sections were deparaffinized, rehydrated and fixed for 20 minutes in 10% neutral-buffered formalin. Antigen retrieval was performed using microwave treatment (MWT) in antigen retrieval solution pH6 or pH9 (AR6 or AR9) according to the target of interest. At each of 4 consecutive staining cycles, primary antibodies (Supplementary Table 4) were added, followed by Opal Polymer HRP Ms + Rb Kit (PerkinElmer) for 10 minutes at room temperature. The tissue sections were then incubated with TSA opal fluorophores (Opal 620, 690, 520 and 540). After each staining cycle, MWT was performed to remove antibody-TSA complex with AR6 or AR9. Finally, all slides were counterstained with DAPI for 5 minutes.

The tissue slides were initially scanned using the PerkinElmer Vectra (v3.0; PerkinElmer) at low magnification (×10). Under pathologist supervision, regions of interest (ROI) were identified and scanned at high resolution (x20) using the Phenochart 1.0.4 viewer (PerkinElmer). High resolution scans of ROIs in the 86 biopsies were analyzed using the software inForm Tissue Finder 2.3.0 (PerkinElmer). The consecutive steps in the image processing software were tissue segmentation based on CD34 (discerning the extravascular from the intravascular compartment), nuclear segmentation based on DAPI staining and cell phenotyping based on CD163, NKp46 and CD3, resulting in a label as "Macrophage", "NK cell", "T cell" and "Other" for each of the identified cells. Infiltration of the respective immune cell types was quantified as density, i.e. number of cell per mm², and relative prevalence, i.e. the number of a specific immune cell type compared to the total number of Macrophages, NK cells and T cells.

### NK cell and monocyte sorting and coculture with glomerular endothelial cells

Peripheral blood mononuclear cells (PBMCs) were isolated from the blood of healthy volunteers from the Etablissement Français du Sang (Besançon, France) by Ficoll gradient centrifugation (Eurobio, Courtaboeuf, France). NK cells were purified from PBMCs by negative selection with magnetic enrichment kit (Stemcell, Grenoble, France) and FcγIII+ CD14- non-classical monocytes were isolated from PBMCs using Slan-(M-DC8)+ Monocyte Isolation Kit, (Miltenyi Biotec, Paris, France)[82]. All sorted cell populations exhibited high purity (>90%), as revealed by flow cytometry.

### Glomerular endothelial cell activation with anti-class I HLA

Glomerular endothelial cells (GENC, Cell System, USA) were cultured in endothelial cell growth medium 2 (PromoCell, Germany) at 37 °C in 5% CO2 until 80% confluence before being used for coculture. GENC were then preincubated with either anti-HLA-A, -B, -C purified antibody (BD Biosciences, France) or control isotype (BD Biosciences) during 30' at room temperature before being washed with phosphate buffered saline. GENC (5.10⁴ cells) were seeded in flat-bottomed 96-well plates and let to adhere 4 h at 37 °C before coculture with immune cells as previously described[34].

### GENC-monocyte coculture

Purified monocytes were mixed with GENC treated with either anti-HLA-A, -B, -C antibody or control isotype at a ratio of 1:1, centrifuged at 100 g for 1 min, and incubated at 37 °C at 5% CO2. After 36 hours of coculture, cells were harvested and Galectin-9 secretion was measured by flow cytometry.

### GENC-NK−monocyte coculture

Purified monocytes and NK cells were mixed with activated GENC at a ratio of 1:1, centrifuged at 100 g for 1 min, and incubated at 37 °C at 5% CO2 in presence or absence of R406, the active metabolite of the Syk inhibitor fostamatinib (2 μM, Invivogen, France). After 36 hours of coculture, supernatants were collected, and stored at −20 °C, until analysis.

## Cytokine assays

For evaluation of intracellular expression of Galectin-9, cells were stained using a fixable viability dye (Fixable Viability Stain 780, BD Biosciences, France) and anti-CD45 BV510 (BD Biosciences) before fixation, permeabilization and staining using an Intracellular staining buffer set (Thermo Fisher Scientific) according to the manufacturer's instructions. Intracellular staining was performed for anti-Galectin-9 FITC (Miltenyi Biotec). Cells were analyzed using a Attune analyzer (Thermo Fisher Scientific).

Human CXCL10, CCL5, and TNF were quantified in culture supernatants using multiplex immunoassay (BD Cytometric Bead Array, BD Biosciences). Data were acquired on Attune analyzer (Thermo Fisher Scientific) and analyzed with FlowJo software (BD Biosciences) using dedicated plugin.

## Human proximal tubule epithelial cell line HK-2

The human proximal tubule epithelial cell line HK-2 (HLA-A2$^{+/+}$) was cultured in DMEM enriched with ITS Liquid Media Supplement (Sigma Aldrich), human EGF (200 ng/mL, Sigma Aldrich), hydrocortisone (500 ng/mL, Sigma Aldrich), triiodothyronin (4 pg/mL, Sigma Aldrich), penicillin/streptomycin (Gibco) and 1% foetal bovine serum (FBS, Dutscher, Brumath, France).

## HLA-A2 specific CD8 + T cells-monocytes coculture

HLA-A2$^{-/-}$ PBMCs were isolated from the blood of healthy volunteers from the Etablissement Français du Sang by Ficoll gradient centrifugation (Eurobio, Courtaboeuf, France). CD8 + T cells were purified by negative selection (magnetic enrichment kit, Miltenyi Biotec) and stained with CellTrace Violet (Thermo Fisher Scientific). Sorted cell population exhibited high purity (>90%), as revealed by flow cytometry. Total monocytes encompassing both FcγRIII+ CD14- non-classical monocytes and FcγRIII- CD14+ classical monocytes were purified by Percoll gradient (Eurobio) centrifugation from the same donor. CD8+ T cells with or without monocytes were co-cultured with 100Gy-irradiated HLA-A2$^{+/+}$ human stimulator cells in X-VIVO 20 medium (Lonza) complemented with 10% human AB serum. Non-specific CD8+ T cells were generated by polyclonal stimulation using CD3/CD28 Dynabeads (1 bead/1cell ratio, Thermo Fisher Scientific) that were magnetically removed after 48 h.

After 4 days of coculture, cells were harvested and CD8 + T cells were either purified by negative selection (magnetic enrichment kit, Miltenyi Biotec) for cytotoxic assay or stained for phenotyping.

For phenotyping, cells were incubated with fixable viability dye (Fixable Viability Stain 780) and stained using anti-CD3 BV785 (BD Biosciences), anti-CD8 PerCP-Cy5.5 (BioLegend), anti-CD14 BV510 (BD Biosciences) and anti-CD107a FITC (Thermo Fisher Scientific) antibodies. The cells were subsequently fixed and permeabilized (Cytofix/Cytoperm fixation/permeabilization kit; BD Biosciences), stained with anti-Granzyme-B AlexaFluor700 (BD biosciences) and anti-IFN-γ PE (BD biosciences) antibodies and analyzed by flow cytometry.

## In vitro cytotoxicity assay – Impedancemetry

In each culture well, 5.10$^3$ HLA-A2$^{+/+}$ HK2 adherent target cells were seeded in E-Plate VIEW 96 PET allowing impedance measurement and incubated at 37 °C in 5% CO$_2$. After 18 h, either HLA-A2 specific or non-specific CD8+ T cells were added to the culture at a 3:1$_{(effector: target)}$ ratio. HK2 viability was monitored every 15 min for 24 h by electrical impedance measurement with an xCELLigence RTCA SP instrument (ACEA Biosciences, San Diego, USA). The cell index was normalized to the reference value (measured just prior to adding effector cells to the culture) of the control wells corresponding to non-specific CD8+ T cells.

## Statistical analysis

We report descriptive statistics using mean and standard deviation (or median and interquartile range for skewed distributions) for continuous variables or numbers, and percentages for discrete variables, for the full cohort and for the rejection subgroups. We used the most recent (as of July 2022) versions of all software programs, including R Studio (version 1.3.1073), SAS (version 9.4, SAS Institute Inc., Cary, NC, United States) and GraphPad Prism (version 9; GraphPad Software, San Diego, CA, United States) for statistical analysis and data presentation.

## Reporting summary

Further information on research design is available in the Nature Portfolio Reporting Summary linked to this article.

## Data availability

All data produced in the present study are available. The Single-cell RNA-sequencing data have been deposited in BioStudies accession code E-MTAB-12051. The images generated by MILAN were made available to the reviewers but are not accessible publicly. These image data can be made available upon request. The kidney transplant biopsy-derived signature matrix encompassing 18 cell types ("KTB18") generated for deconvolution is available in Supplementary Data 1. This Signature matrix file can be directly used as custom input to run a job within the CIBERSORTx console (https://cibersortx.stanford.edu/runcibersortx.php). Source data are provided with this paper.

## Code availability

Code for Seurat is available at https://satijalab.org/seurat/. Code for CellChat is available at https://github.com/sqjin/CellChat. Code for Slingshot is available at https://github.com/kstreet13/slingshot.

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

## Acknowledgements

We thank the clinicians and surgeons, nursing staff and the patients of University Hospitals Leuven. We thank the clinical centers of the BIO-MARGIN consortium, the clinicians, surgeons, nursing staff and patients. Seventh Framework Program (FP7) of the European Commission, in the HEALTH.2012.1.4-1 theme of "innovative approaches to solid organ transplantation" (grant agreement no. 305499). Research Foundation Flanders (F.W.O.), Belgium, under the frame of ERACoSysMed-2, the European Research Area Network for Systems Medicine in Clinical Research and Medical Practice (project ROCKET, JTC2-29), and with a project grant (grant agreement G0087620N). JC, EVL and TV are supported by a PhD Fellowship grant from the Research Foundation Flanders (F.W.O.) (grant n° 1196119 N, 1143919 N, and 1S93918N, respectively). MN and BS are supported by the Research Foundation Flanders (F.W.O.) as senior clinical investigator (grant agreement no. 1844019 N and 1842919 N, respectively). BL is supported by the MiMedI project funded by BPI France (grant n°DOS0060162/00), the European Union through the European Regional Development Fund of the Region Bourgogne Franche-Comté (grant n°FC0013440) and the Agence Nationale de la Recherche (grant JCJC n°ANR-22-CE18-0011-01).

## Author contributions

B.L., J.C. and M.N. conceived and designed the study. B.L., J.C., Y.V.H., A.A., F.D.S., P.K., F.B. and M.N. validated the MILAN multiplex immunohistochemistry platform for kidney biopsies, performed the analyses and interpreted the MILAN data. M.E. and M.R. performed the OPAL analyses. B.L., J.C., P.B., T.D., A.F., D.L., M.M.S., T.K.S., C.T., T.V.B., E.V.L. and M.N. generated, analysed and interpreted the single cell and bulk transcriptomic data. J.U.B., P.K. and M.R. interpreted the biopsy histology. B.L., V.M. and O.T. performed the in vitro experiments. J.C., D.A., K.D.V., W.G., D.K., P.M., A.S., B.S., C.T., A.V.C., E.V.L. T.V. and M.N. included patients and collected and interpreted the clinical data. B.L., J.C. and M.N. wrote the paper with contribution from all co-authors.

## Competing interests

The authors declare no competing interests.

## Additional information

[1]Department of Microbiology, Immunology and Transplantation, Nephrology and Kidney Transplantation Research Group, KU Leuven, Leuven, Belgium. [2]Université de Franche-Comté, UBFC, EFS, Inserm UMR RIGHT, Besançon, France. [3]Department of Nephrology and Kidney Transplantation, University Hospitals Leuven, Leuven, Belgium. [4]Department of Oncology, Laboratory for Experimental Oncology, KU Leuven, Leuven, Belgium. [5]Department of Imaging and Pathology, Translational Cell and Tissue Research, KU Leuven, Leuven, Belgium. [6]Department of Nephrology and Kidney Transplantation, Necker-Enfants Malades Hospital, Assistance Publique-Hôpitaux de Paris, Paris, France. [7]Université Paris Cité, Inserm U1151, Necker Enfants-Malades Institute, Paris, France. [8]Division of Multi-Organ Transplantation, Department of Surgery, UCSF, 513 Parnassus, San Francisco, CA, USA. [9]Institute of Pathology, University Hospital Cologne, Cologne, Germany. [10]VIB Center for Cancer Biology, Leuven, Belgium. [11]Department of Human Genetics, Laboratory of Translational Genetics, KU Leuven, Leuven, Belgium. [12]Department of Nephrology, Hannover Medical School, Hannover, Germany. [13]Department of Imaging and Pathology, KU Leuven, Leuven, Belgium. [14]Department of Pharmacology and Transplantation, University of Limoges, Inserm U1248, Limoges University Hospital, Limoges, France. [15]EFS, HLA Laboratory, Décines, France. [16]Université Claude Bernard Lyon I, Inserm U1111, CNRS UMR5308, CIRI, Ecole Normale Supérieure de Lyon, Lyon, France. [17]Department of Pathology, Necker-Enfants Malades Hospital, Assistance Publique-Hôpitaux de Paris, Paris, France. [18]Histocompatibility and Immunogenetics Laboratory, Red Cross-Flanders, Mechelen, Belgium. [19]Hospices Civils de Lyon, Edouard Herriot Hospital, Department of Transplantation, Nephrology and Clinical Immunology, Lyon, France. [20]Department of Nephrology and Kidney Transplantation, Dijon Hospital, Dijon, France. [21]These authors contributed equally: Baptiste Lamarthée, Jasper Callemeyn, Yannick Van Herck, Asier Antoranz. ✉e-mail: maarten.naesens@uzleuven.be

