## [Peer Review File · Nature Communications]

Transcriptional and spatial profiling of the kidney allograft unravels a central role for FcyRIII+ innate immune cells in rejectionREVIEWER COMMENTS

Reviewer #1 (expertise in spatial transcriptomics, kidney transcriptomics):

This manuscript by Lamarthee et. al seeks to uncover the interactions between alloreactive inflammatory infiltrate. The investigators utilize a variety of orthogonal techniques to characterize immune cell infiltration. These techniques include scRNAseq, MILAN-multiplexed immunofluorescence, and bulk RNAseq with deconvolution. The paper is well written with advanced analyses provided for the scRNAseq and MILAN immunofluorescence datasets. However, enthusiasm is somewhat tempered by a lack of novelty in the study's findings.

The authors show the following:

1. A subset of immune and epithelial cells of the kidney are identified by scRNAseq
2. Using X and Y chromosome genes, the authors illustrate that epithelial cells are innate to the donor and immune cells are (mostly) innate to the recipient.
3. There are an increased number of NK and myeloid cells in the setting of graft inflammation as determined by the scRNAseq, bulk RNAseq, and MILAN (protein) datasets.
4. FCGR3A+ NK and myeloid cells are associated with vascular inflammation in ABMR.
5. There are differentially expressed genes and evidence of cell-cell communication in FCG#3A+ NK and myeloid cells in the setting of rejection.

Main critiques:

1. Novel findings could be better emphasized - There are dozens of manuscripts illustrating the role of FCGR3A+ NK cells in ABMR and endothelial damage. This study is a validation and further confirms its predecessors by using scRNAseq and MILAN. Can the authors better emphasize a novel finding from this study? Despite the lack of novelty, this manuscript has the potential to shine as a transplant atlas, perhaps molecularly annotating features of the Banff criteria. However, the small sample size, lack of disease variety, and lack of epithelial cell annotation currently reduce this study's utility as a benchmark atlas.
2. Clinicopathological diagnoses of acute rejection subtypes are not consistent - The authors no longer use their former rejection classification criteria. In their 2021 JASN article (JASN 32: 1084-1096, 2021), the authors reclassify Banff rejection in 3510 samples into 6 novel clusters based on unsupervised clustering of histopathologic features. Three clusters were DSA+ and three were DSA- with parallel clusters for no rejection, tubulointerstitial inflammation, and glomerular inflammation (Figure 1 of the JASN publication). In this study's methods, the authors state all biopsies were "categorized into one of six recently described clinicopathological clusters" (Line 144) and cite their JASN study.
 - a. However, Fig 1f does not cluster samples according to these clusters and instead annotates 4 different groups. How were these 4 groups chosen/defined?
 - b. How do the 16 biopsies fit into the prior 6 clusters?
Would the 16 samples maintain the same disease classification in the prior clustering approach? (for example, given the radius and theta location, would biopsy #8 be classified as borderline in Figure 2B of the JASN study? Would biopsy #16 be classified as mixed rejection?).
 - c. In figure 2 and 3 of the present study, the authors change categorization again, now having 6 clusters which are different than those of Figure 1 or of the clusters from the JASN paper referenced in their methods.
 - d. There are benefits to unsupervised clustering, but if all analyses are internally normalized (or re-clustered) for major disease definitions, it reduces the ability of nephrologists and nephropathologists to interpret your data, or translate the molecular categorization into practice. Can the authors provide a ground truth that a nephrologist or nephropathologist could utilize to understand these results?
3. The sample size is small- Although the authors characterize 35,152 single cells in 16 biopsies from 14 individuals, only 1 individual had TCMR and 3 had ABMR. It is recommended that the authors improve the balance of no rejection and TCMR/ABMR samples within the dataset. In addition, multiple important epithelial cells of the kidney remain unannotated or under-represented. For example, only 159 podocytes are identified and some samples have as few as 1 podocyte (#16). There are no clusters for the DCT or principal cells. The proximal tubule is treated as a single cluster and no cell-cell communication was found. If the resolution is increased, PT injury clusters may manifest which are targets of immune cell signaling.

4. Epithelial/Endothelial-immune cell communication is not discussed – With the focus on immune cell communication, I would be curious to see the authors evaluate immune-epithelial interactions to a greater extent. The Banff criteria emphasizes tubulitis, interstitial inflammation, and glomerulitis. The cell-cell communications underlying these criteria are not evaluated. Are there injury clusters of podocytes and tubules? Although the authors localize the FCGR3A+ cells to the vasculature, the interactions with the endothelium are minimally explored. Are there inflamed endothelial sub-clusters? Are their stromal cells or fibroblasts that interact with FCGR3A+ cells?

5. Data availability – I could not access the raw data for review. E-MTAB-12051 is not publicly available. The MILAN images are not available for review to evaluate carryover staining between markers. This is important because 26% of cells remain uncharacterized.

Specific critiques:

6. Supp Table 1 –Please indicate which biopsies were repeats.
7. Figure 3D – If most of the cells in snRNAseq are PT cells (>50%), why do only 6.93% label with AQP1?
8. Why are most epithelial markers eliminated from Supp Table 3 ?
9. Fig 4B –after the meticulous efforts to characterize monocyte sub-types, the “NOS monocytes” were most significantly associated with inflammation. Can these cells be better understood?
10. Supp Fig 3B – would suggest eliminating podocytes from the dot plot since there is only 1 podocyte in that analysis.
11. Fig 9 – why are there so few endothelial-immune cell interactions given the findings of vascular inflammation in ABMR?

Reviewer #2 (expertise in nephrology, Fc receptors in kidney):

The manuscript by Lamarthée et al reports a central involvement of innate immune cells, notably FcγRIII+ NK and FcγRIII+ nonclassical monocytes, in the pathogenesis of allograft rejection suggesting several potential therapeutic targets to improve allografts. They use droplet-based single-cell RNA sequencing and described 38 markers new markers on 18 independent biopsy slides. Their data strongly pointed for the role of FcγRIII+ NK and FcγRIII+ nonclassical monocytes in antibody-mediated rejection, with specificity to the glomerular area.

Comments:

As discussed by the authors I agree that ‘the relation between the histological/molecular phenotype and cellular mix in the biopsies, and the underlying causal pathways of allorecognition or even autoreactivity was not assessed’ and will need further study.

Nevertheless, to obtain indirect confirmation such hypothesis, the authors may address with in vitro experiments using blood NK and monocytes from those patients to show antibody-mediated cytotoxicity (ADCC or ADC). This will support the proposed mechanisms in allograft rejection.

Reviewer #3 (expertise in nephrology, kidney transplantation):

This paper by Lamarthée et al., examined the transcriptional and spatial profiling of the kidney allograft exploring the role of FcγRIII+ innate immune cells in mediating allograft rejection in human kidney transplant biopsies. The authors performed single-cell RNA-sequencing of 35,152 transcriptomes from 16 kidney transplant biopsies with varying rejection diagnoses and intensity of rejection these were compared to data derived from biopsies without rejection and generated cell-type specific gene expression signatures. The authors identified a specific association between recipient-derived FCGR3A+ monocytes, FCGR3A+ NK cells and the severity of allograft inflammation. Activated FCGR3A+ monocytes overexpressed CD47 and lymphocyte immunoglobulin-like receptor (LILRA) genes and increased signaling pathways that initiated T cell infiltration. FCGR3A+ NK cells overexpressed FCRL3, suggesting that antibody-dependent cytotoxic activity is a central mechanism of NK cell-mediated graft injury. Multiplexed immunofluorescence using 38 markers on 18 independent biopsy slides confirmed this role of FcγRIII+ NK and FcγRIII+ nonclassical monocytes in antibody-mediated rejection, with specificity to the glomerular area.

Critique: Overall, this is a well written and impressive manuscript that will likely change the way

that clinicians look at well-established paradigms of allograft rejection, which focus primarily on adaptive immunity and T-cell activation. These investigators show that the primary initiating events and propagation are associated with innate immunity involving FcγR3A+ NK cells and FcγR3A + monocytes. This is true for both antibody-mediated rejection (i.e., ADCC) and cell-mediated rejection where innate immune events mediated by LILRA molecules on FcγR3A+ monocytes show that these cells are not only capable of recognizing IgG binding to target cells in the allograft but also have the capacity to directly recognize non-self on donor cells through the LILRA pathway, initiating CD8+ T-cell activation. The same is true for FcγR3A+ NK cells where KIR (killer inhibitor receptor) can initiate direct recognition of non-self HLA and drive CD8+T cell activation.

Importantly, one must consider the implications of these findings for clinical therapeutics. The authors conclude: "We have demonstrated the importance of FcγRIII+ monocytes and FcγRIII+ NK cells, and innate immune crosstalk during rejection. We believe that this work is a major resource for understanding of the role of these cells and the innate immune system involvement in kidney transplant rejection and shows promise for developing therapeutic strategies that target these pathways." This is the essence and productive culmination of this type of work. In this regard, there have been some attempts to address this. Shin et al. (Shin BH, Ge S, Mirocha J, Karasyov A, Vo A, Jordan SC, Toyoda M. Regulation of anti-HLA antibody-dependent natural killer cell activation by immunosuppressive agents. *Transplantation*. 2014 Feb 15;97(3):294-300. doi: 10.1097/01.TP.0000438636.52085.50. PMID: 24342979.) showed using an in vitro analysis of ADCC looking at CD56+ NK cell activation that steroids and CNI were the most effective agents in reducing NK+ cell γ -IFN release in ADCC. The authors mention there may be promise for mTOR inhibitors, but in the study by Shin et al., no effect was seen. I think it would be important for the authors to also reference emerging data from the use of daratumumab (anti-CD38) which is used to treat multiple myeloma. CD38 is expressed on multiple cell types including NK cells and monocytes. Recent data from case studies (Doberer K, Kläger J, Gualdoni GA, Mayer KA, Eskandary F, Farkash EA, Agis H, Reiter T, Reindl-Schwaighofer R, Wahrmann M, Cohen G, Haslacher H, Bond G, Simonitsch-Klupp I, Halloran PF, Böhmig GA. CD38 Antibody Daratumumab for the Treatment of Chronic Active Antibody-mediated Kidney Allograft Rejection. *Transplantation*. 2021 Feb 1;105(2):451-457. doi: 10.1097/TP.0000000000003247.) showed that daratumumab can be effective in treating antibody-mediated rejection, even when donor antibodies are not reduced. This suggests, possible depletion of NK cells and monocytes in the glomeruli capillary loops. Unfortunately, CD38 is also expressed on Treg cells and treatment can initiate CMR episodes. I think a mention of these studies would be helpful in directing discussion of possible future therapies based on their findings.

RESPONSE TO REVIEWER COMMENTS

Reviewer #1 (expertise in spatial transcriptomics, kidney transcriptomics):

This manuscript by Lamarthee et. al seeks to uncover the interactions between alloreactive inflammatory infiltrate. The investigators utilize a variety of orthogonal techniques to characterize immune cell infiltration. These techniques include scRNAseq, MILAN-multiplexed immunofluorescence, and bulk RNAseq with deconvolution. The paper is well written with advanced analyses provided for the scRNAseq and MILAN immunofluorescence datasets. However, enthusiasm is somewhat tempered by a lack of novelty in the study's findings.

The authors show the following:

1. A subset of immune and epithelial cells of the kidney are identified by scRNAseq
2. Using X and Y chromosome genes, the authors illustrate that epithelial cells are innate to the donor and immune cells are (mostly) innate to the recipient.
3. There are an increased number of NK and myeloid cells in the setting of graft inflammation as determined by the scRNAseq, bulk RNAseq, and MILAN (protein) datasets.
4. FCGR3A+ NK and myeloid cells are associated with vascular inflammation in ABMR.
5. There are differentially expressed genes and evidence of cell-cell communication in FCGR3A+ NK and myeloid cells in the setting of rejection.

We thank the reviewer for the detailed evaluation of our manuscript, the positive feedback and constructive suggestions.

Main critiques:

1. Novel findings could be better emphasized - There are dozens of manuscripts illustrating the role of FCGR3A+ NK cells in ABMR and endothelial damage. This study is a validation and further confirms its predecessors by using scRNAseq and MILAN. Can the authors better emphasize a novel finding from this study? Despite the lack of novelty, this manuscript has the potential to shine as a transplant atlas, perhaps molecularly annotating features of the Banff criteria. However, the small sample size, lack of disease variety, and lack of epithelial cell annotation currently reduce this study's utility as a benchmark atlas.

We thank the reviewer for this comment and agree that our sample set for the scRNAseq studies is relatively limited (N=16). We do not claim that these 16 biopsies are representative of all possible phenotypes of kidney transplant pathology. Nevertheless, we believe that all datasets used in this study (not only the scRNAseq data) are relevant for use by other groups, through unlimited access to the source data. Furthermore, in the revised manuscript, we have avoided using the term "atlas", to not suggest comprehensiveness in terms of phenotypic spectrum (lines 148 and 356).

The results of the present study suggest that innate immune activation is at least equally relevant, if not more so, in human kidney transplant rejection than adaptive immune responses. This is highlighted as the primary conclusion of our manuscript, both at the end of the abstract and at the end of the discussion section.

Several previous manuscripts indeed suggested FcγRIII+ NK cells' role in ABMR pathophysiology. However, the phenotype of these cells at both single cell transcriptomic and proteomic levels within the kidney transplant allograft and in rejection context remained ill-defined. Using scRNAseq on 16 biopsies, multiplex staining on 18 independent biopsies, and deconvolution of 200 bulk transcriptomes in two cohorts from different clinic centers, we show that the fraction of infiltrating

FcγRIII+ NK cells in the allograft is directly correlated to the inflammation severity and that the NK cells present a profile of ADCC activation. To our knowledge, this is a first study to demonstrate that this NK cells burden is associated with inflammation severity with such robustness.

Our results are even more novel regarding FcγRIII+ nonclassical monocytes. For the first time and with highly robust and cutting edge methods, we showed that FcγRIII+ nonclassical monocytes are mainly recipient-derived, that the burden of these cells is also associated with rejection severity regardless of ABMR or TCMR categories and that these cells express LILR as well as FcγRIII. This enables these cells to recognize donor's tissue, independent of adaptive immunity or antibody recognition. In addition, we showed that these cells interact with all other cells and express CXCL10 in the allograft making them potential targets for new therapeutic approaches. Given the increasing interest in the role of these myeloid cells in the pathogenesis of allograft rejection (currently confined to *in vivo* studies), our human data will contribute importantly to the knowledge on this topic.

We do not follow the argument that our study is hampered by small sample size and lack of disease variety. We gathered 16 fresh kidney allograft biopsies for scRNASeq, which to our knowledge is by far the largest cohort in this field. The number of cells is necessarily limited using needle biopsies compared to nephrectomy samples, but using biopsies in functioning organs permits to investigate actual biologically relevant samples. More importantly, we validated our findings using deconvolution in hundreds of samples from multiple centers and we used 18 independent biopsies to generate more than 500 000 proteomic single cell profiles using MILAN multiplex technique. In terms of disease variety, we agree that our scRNAseq cohort presented less phenotypic heterogeneity than our other cohorts. For instance, no fresh biopsy from the scRNAseq cohort presented mixed rejection (Figure 1f). However, this phenotype was represented in MILAN's cohort (Figure 3e), and the latter analysis confirmed the role of the FcγRIII+ cells in this phenotype. Finally, we validated our findings in independent scRNAseq datasets, now also encompassing mixed rejection cases (new Suppl Fig 6), and also using the deconvolution strategy. As illustrated in figure 2b, the first deconvolution cohort (GSE147089) presented a broad range of inflammation severity and histological lesions or DSA status as depicted by the dispersion of the dots on the polar plot.

Finally, we consciously focused our analyses on immune compartment characterization. We agree that the non-immune response by structural cells (including epithelial cells), upon kidney rejection is also crucial. We are currently preparing another full-length manuscript to extensively address this issue. Unless the editor considers it important to also integrate detailed analyses of the structural cell responses to inflammation in the current manuscript, we suggest to not further overload the current manuscript by adding these data to the paper.

2. Clinicopathological diagnoses of acute rejection subtypes are not consistent – The authors no longer use their former rejection classification criteria. In their 2021 JASN article (JASN 32: 1084–1096, 2021), the authors reclassify Banff rejection in 3510 samples into 6 novel clusters based on unsupervised clustering of histopathologic features. Three clusters were DSA+ and three were DSA- with parallel clusters for no rejection, tubulointerstitial inflammation, and glomerular inflammation (Figure 1 of the JASN publication). In this study's methods, the authors state all biopsies were "categorized into one of six recently described clinicopathological clusters" (Line 144) and cite their JASN study.

a. However, Fig 1f does not cluster samples according to these clusters and instead annotates 4 different groups. How were these 4 groups chosen/defined?

b. How do the 16 biopsies fit into the prior 6 clusters?

Would the 16 samples maintain the same disease classification in the prior clustering approach? (for example, given the radius and theta location, would biopsy #8 be classified as borderline in Figure 2B of the JASN study? Would biopsy #16 be classified as mixed rejection?).

c. In figure 2 and 3 of the present study, the authors change categorization again, now having 6 clusters which are different than those of Figure 1 or of the clusters from the JASN paper referenced in their methods.

We are sorry for this misunderstanding. Actually, our classification is consistent in all our figures (1f, 2b and 3e). We want to draw Reviewer 1's attention on the fact that the biopsies used for scRNAseq (N=16) and depicted in Figure 1f and Supp Table 1 are strictly independent of the biopsies used for MILAN multiplex staining (N=18) presented in Figure 3e and Supp Table 2 as well stipulated in the Material and Methods section: *"For the multiplex immunofluorescence (MILAN) analysis, an independent set of 18 biopsies was included from renal transplant recipients followed in the University Hospitals Leuven, Belgium (Supplementary Table 2)"*.

This is why the polar plots represented in Figures 1f and 3f display different dispersion of the dots and thus different classifications. In the scRNAseq cohort, none of the 16 fresh biopsies clustered with Mixed rejection or ABMRh DSA- classifications whereas certain biopsies used for MILAN multiplex staining were classified in these categories using the semisupervised classification we previously published (Vaulet et al, JASN 2021). We have now modified Figure 1f, Figure 2b and 3e in order to present a consistent legend and added several sentences in the result section to clarify this:

Results Page14 Lines 396-400: *"We first stratified the biopsies using a data-driven clustering method producing a phenotypic reclassification of kidney transplant rejection¹⁹. As depicted on the polar plot generated by this method, biopsies were categorized in four different groups: non-rejection (NR) DSA negative group (NR DSA-,350 N=4), TCMR (N=1), NR DSA positive (N=8) and ABMR DSA+ (N=3). No biopsy in this cohort was categorized as mixed TCMR-ABMR or DSA- histology of ABMR (ABMRh)."*

Results Pages 14-15 Lines 407-420: *"In this aim, we generated a kidney transplant biopsy-derived signature matrix encompassing 18 cell types (KTB18) and estimated the cellular fractions (Fig.2a-b)¹⁷... We next applied the same data-driven clustering method as used for the scRNASeq cohort¹⁹ on the bulk transcriptomics dataset (N=224). This indicated a good distribution of the biopsies on the polar plot, covering the entire spectrum of rejection phenotypes (Fig.2b). We observed that the cell types best associated with the severity of inflammation were nonclassical monocytes and NK cells, as well as classical monocytes, macrophages and various CD8+ T cell subsets (Fig.2c)."*

d. There are benefits to unsupervised clustering, but if all analyses are internally normalized (or re-clustered) for major disease definitions, it reduces the ability of nephrologists and nephropathologists to interpret your data, or translate the molecular categorization into practice. Can the authors provide a ground truth that a nephrologist or nephropathologist could utilize to understand these results?

We thank this reviewer for this important comment. A main problem with the current clinical classification ("Banff classification") is the arbitrary rules that are applied to the cases, which attributes sometimes non-inflamed cases to rejection classes while other overtly inflamed biopsies do not fulfill the strict criteria. Although the Banff classification has its place in clinical context and in large epidemiological studies, it is performing much less in smaller-sized studies like the scRNASeq (N=16) or MILAN (N=18) cohort. Cases that are phenotypically very similar could be classified very differently according to the Banff system, which leads to dramatic decreases in the statistical power to detect differences between categories. For consistency, we also classified the N=224 bulk transcriptomic dataset according to the same system.

The semisupervised classification we used here was extensively discussed in Vaulet et al JASN 2021, and the clinical relevance and external validity was demonstrated in that paper. This classification categorizes the biopsies according to the severity of the rejection and is directly associated to the graft survival. We used the publicly available online tool developed by our team concomitant to this previous publication to classify our biopsies accordingly: <https://rejectionclass.eu.pythonanywhere.com>. There was no internal normalization or reclustering. We simply renamed the 6 clusters that were previously identified as Cluster 1, 2, 3, 4, 5, and 6 in the previous article (Figure 2b of the JASN article) as No Rejection DSA-, No Rejection DSA+, TCMR, ABMRh DSA-, ABMR DSA+ and Mixed Rejection respectively in order to help the readers (especially nephrologist and nephropathologist) to match these new clusters with diseases/.../terminology...phenotypes the field is used to.

Finally, our evaluation of inflammation severity aligns with the ambition of the Banff classification system to integrate such more granular scoring systems in its definitions. This was discussed at the Banff'22 meeting in September 2022 (Banff report in preparation). Although it is currently too early to integrate this approach in clinical routine, this data-driven instead of rules-based approach is identified as a way forward for the field, and further validation studies are initiated. Beyond the conclusions made on the pathobiology of rejection, the current manuscript will also contribute to these discussions. We also kindly refer to the editorial commentary by Vasquez-Rios and Menon on this topic: <https://doi.org/10.1681/ASN.2021030348>

3. The sample size is small— Although the authors characterize 35,152 single cells in 16 biopsies from 14 individuals, only 1 individual had TCMR and 3 had ABMR. It is recommended that the authors improve the balance of no rejection and TCMR/ABMR samples within the dataset. In addition, multiple important epithelial cells of the kidney remain unannotated or under-represented. For example, only 159 podocytes are identified and some samples have as few as 1 podocyte (#16). There are no clusters for the DCT or principal cells. The proximal tubule is treated as a single cluster and no cell-cell communication was found. If the resolution is increased, PT injury clusters may manifest which are targets of immune cell signaling.

We admit that 35,152 cells for the scRNAseq analysis could be considered as a limited number but here we gathered 16 fresh biopsies which to our knowledge is the biggest cohort in term of human kidney allograft biopsies. Given the limited number of patients with ABMR or TCMR diagnostic, it took the team 2 years to gather all these fresh and adequate biopsies. More importantly, we validate our findings in another scRNAseq cohort (new figure S6, lines 522-533), with the addition of cases with mixed AMR and TCMR and we confirmed our transcriptomic-based findings using deconvolution in two independent cohorts from different clinical centers (N=224 patients and N=403 patients respectively). We performed multiplex staining with two different techniques (OPAL and MILAN) on two other independent cohorts (N=90 patients and N=18 patients respectively) in order to conduct a more biologically meaningful analysis and notably to correlate transcriptomic findings with ground truth proteomic-based observations. In this aim, more than 500 000 cells were analyzed using the MILAN method to confirm the importance of FcγRIIIa+ cells as well as their actual location in the allograft. We hope that the reviewer and editor follow us in the reasoning that all these analyses add to the robustness of our conclusions and that the study's relevance is not limited to the scRNAseq data.

Regarding epithelial cells, we completely agree with the reviewer. As previously discussed, we consciously focused our analyses on immune compartment characterization. As illustrated in figure S1a, with a resolution=2, we observed 14 clusters we considered as PT cells. Of course, this cluster

encompasses PT cells from different segments such as PTS1, S2 and S3 as previously reported in human (Lake et al, Nat Commun 2019) and in mouse (Lu et al, JASN, 2021, 10.1681/ASN.2020081143). The interactions with immune compartment could be different according to these segments and according to these cell types (see below, and also new Figure 10). In addition, given the complexity of the structural cell responses to the inflammatory milieu upon rejection, we are currently assessing the responses by the non-immune compartment in a fully dedicated manuscript. As stated above, if the editor considers it necessary to further expand the current manuscript in this direction, we are willing to do so, but we are reluctant because it would make the message more diffuse.

4. Epithelial/Endothelial-immune cell communication is not discussed – With the focus on immune cell communication, I would be curious to see the authors evaluate immune-epithelial interactions to a greater extent. The Banff criteria emphasizes tubulitis, interstitial inflammation, and glomerulitis. The cell-cell communications underlying these criteria are not evaluated. Are there injury clusters of podocytes and tubules? Although the authors localize the FCGR3A+ cells to the vasculature, the interactions with the endothelium are minimally explored. Are there inflamed endothelial sub-clusters? Are their stromal cells or fibroblasts that interact with FCGR3A+ cells?

This is a very relevant remark. We reintegrated PT cells, endothelial cells, podocytes and immune cells of interest (*FCGR3A*+ NK and *FCGR3A*+ monocytes) to better characterize immune-epithelial/endothelial interactions as suggested by this Reviewer. Please find a new figure 10 depicting these results. We did not identify any injury clusters of podocytes given the limited number of these cells. However, we distinguished 4 different clusters within PT cells corresponding to the 3 segments already described from kidney cortical area to medulla as previously described by Lake et al Nature Communications 2019 (PT_S1, PT_S2 and PT-S3 respectively, Fig.10a,c). Interestingly, a cluster corresponding to injured tubular cells expressing *HAVCR1* and *VCAM1* (Fig.10b) was detected as previously described (Rinaldi et al, JCI Insight, 2022, 10.1172/jci.insight.161783). Regarding endothelial cells, in addition to the previously described clusters (ECptc, ECg and ECvr), we detected a cluster expressing high level of *VCAM1* suggesting an activation as previously reported (Malone et al, JASN, 2021).

Next, we performed CellChat analysis assessing both secreted signaling and cell-cell contact using *FCGR3A*+ NK and *FCGR3A*+ monocytes as source cells. As depicted in novel Fig.10d depicted below, very limited cellular communications were detected in PT cells clusters which is in line with Fig.10e confirming that no significant interaction was detected in the tubules. One could speculate that tubular injury is indirect and mainly results from a metabolic shutdown associated with the vascular rejection. In contrast, *FCGR3A*+ NK and *FCGR3A*+ monocytes mainly interact with endothelial cells. More specifically, *VCAM1*+ EC (EC_Injured) overexpress *ACKR1* and *TNFRSF1A* making them potential

targets of monocytes-derived CXCL10 and TNF but also of NK-derived CCL5.

Results Page 21 Lines 598-612: “To evaluate epithelial/endothelial-immune cell communications to a greater extent, we reintegrated PT cells, endothelial cells, podocytes and immune cells of interest: FCGR3A+ NK and FCGR3A+ myeloid cells (Fig. 10a). We did not identify any injury cluster of podocytes given their limited number. However, four different clusters were distinguished within PT cells corresponding to the 3 segments already described from kidney cortical area to medulla (PT_S1, PT_S2 and PT-S3 respectively, Fig. 10a-c). Interestingly, a cluster corresponding to injured tubular cells expressing HAVCR1 and VCAM1 (Fig. 10b) was detected⁵⁶. Regarding endothelial cells, in addition to the previously described clusters (ECptc, ECg and ECvr), we detected a cluster expressing high levels of VCAM1, suggesting an activation as previously reported⁴⁷. We performed CellChat analysis assessing both secreted signaling and cell-cell contact using FCGR3A+ NK and FCGR3A+ monocytes as source cells. Very limited cellular communications were detected in PT cells clusters (Fig. 10d), which is in line with Fig. 10e confirming that no significant interaction was detected between the tubules and FCGR3A+ immune cells. One could speculate that tubular injury is indirect and mainly results from a metabolic shutdown associated with the vascular rejection. In contrast, secreted signaling unraveled that FCGR3A+ monocytes mainly interact with NK cells through LGALS9 coding for galectin 9 inferred secretion (Fig. 10e).”

We previously showed that anti-HLA DSA triggers such cytokine production in kidney transplant recipients' serum independent of histological lesions and we demonstrated that this secretion could be induced by FcγRIII+ cells in an *in vitro* model of ADCC mimicking anti-HLA DSA binding on endothelium (Van Loon et al, Front Immunol, 2022, 10.3389/fimmu.2022.818569). Using the exact same *in vitro* model, we show in a new figure S9 depicted below that Syk inhibition by R406, the active form of fostamatinib reduces this secreted signaling. Regarding cell-cell contact, the main interactions were driven by *ITGB2* and *ITGA4* expression in both FCGR3A+ NK and FCGR3A+ monocytes. The basal expression of their ligands *ICAM1* and *ICAM2* in ECvr, ECptc and ECg strengthened by *VCAM1* overexpression in EC_Injured could explain this preferential cell-cell contact.

Suppl Fig. 9

Results Page 22 Lines 612-630: *“We previously showed that anti-HLA DSA triggers chemokine and cytokine production in kidney transplant recipients’ serum, independent of histological lesions and we demonstrated that this secretion could be induced by FcγRIII+ cells in an in vitro model of ADCC mimicking anti-HLA DSA binding on endothelium²². Here, using the same in vitro model, we found that non-classical monocytes significantly secreted more galectin 9 after anti-HLA DSA recognition (Fig.S8a-d). Therefore, we measured the expression of Tim-3, the galectin 9-receptor in FcγRIII+ NK cells in kidney biopsies using multiplexed immunofluorescence (MILAN). We observed that Tim-3 expression is increased in FcγRIII+ NK cells closely located near FcγRIII+ monocytes in acute rejection context (Fig.S8e-f) suggesting that galectin-9 secreted by FcγRIII+ monocytes could trigger NK cells activation in rejection context.*

In addition, FCGR3A+ NK and FCGR3A+ monocytes interact with endothelial cells. More specifically, VCAM1+ EC (EC_Injured) overexpress ACKR1 and TNFRSF1A making them potential targets of monocytes-derived CXCL10 and TNF but also of NK-derived CCL5. Given that the recognition of a Fc fragment by the FcγRIII receptor induce Syk dependent pathways triggering cytokine secretion, we investigated whether a Syk inhibitor blocks the secreted signaling using the exact same in vitro model. The addition of R406, the active form of fostamatinib, during the NK/non-classical monocytes/GENC coculture completely abrogated the secretion of TNF, CCL5 and CXCL10 in presence of HLA DSA, suggesting that Syk inhibitors could alleviate the cytokine release of NK cells and monocytes upon ABMR (Fig.S9a-b).”

5. Data availability – I could not access the raw data for review. **E-MTAB-12051 is not publicly available. The MILAN images are not available for review** to evaluate carryover staining between markers. This is important because 26% of cells remain uncharacterized.

Please find the reviewer login (view only) corresponding to E-MTAB-12051:

Reviewers login (view only)

Username: Reviewer_E-MTAB-12051

Password: KF9Vvmkw

The scRNAseq data will stay private until publication.

The MILAN platform is very novel. We have worked very hard on creating a viewer. Please find below the link toward the viewer:

<https://kid.milan-tool.com>

Username: reviewer@kid.milan-tool.com

Password: S8Ylt36!

Specific critiques:

6. Supp Table 1 –Please indicate which biopsies were repeats.

As requested, we have indicated in the Supp Table 1 the biopsies derived from the same patients in the revised version of the manuscript.

7. Figure 3D – If most of the cells in snRNAseq are PT cells (>50%), why do only 6.93% label with AQP1?

These two percentages cannot be directly compared as suggested by Reviewer 1. They are derived from two different technologies assessing two different entities (gene expression vs protein level) from two independent sets of biopsies.

In addition, in figure 1, we chose to pool all the 14 clusters we detected with a resolution of 2 (Figure S1a) and which express specific tubular markers (*CUBN*, *MIOX*, *LRP2*, *CDH6*, Figure1d) as previously described at single cell transcriptomic level in human kidney (Lake et al, Nat Commun, 2019, PMID 31249312). We identified this pool of clusters representing >50% of all the cells as “PT cells”. In contrast, *AQP1* gene expression is more endothelial-related in our dataset (please, see figure below) which is in line with Lake et al Nature Communications 2019 report. They also suggested that *AQP1* gene expression can be useful for PT subset in order to distinguish the PT-specific segment S3 which is located near the medulla.

Consistently, AQP1 protein is only detected in 6.93% of tubule cells using MILAN multiplex technology as illustrated in figure 3b. More precisely, we observed a continuum of expression of AQP1, CD138 and PanCK at protein level among all the tubuli, from proximal tubules to distal tubules which is consistent with gene expression.

8. Why are most epithelial markers eliminated from Supp Table 3 ?

The epithelial markers that we have included in the panel are: AQP1, CD138, PanCK, and S100. With these 4 epithelial markers we were able to discriminate 4 different types of epithelial cells: proximal tubuli (PanCK+,S100+), distal tubuli (AQP1+,CD138+), CD138+ tubuli, and AQP1+PanCK+ tubuli. This subpanel did not allow us to identify other epithelial cells such as loop of Henle cells. To distinguish the epithelial compartment from the interstitium, we have trained a pixel classifier using QuPath based also on these markers (see methods, MILAN in-silico microdissection and neighborhood analysis).

9. Fig 4B –after the meticulous efforts to characterize monocyte sub-types, the “NOS monocytes” were most significantly associated with inflammation. Can these cells be better understood?

MILAN is not like single-cell RNA-seq, it requires the pre-selection of a protein panel. This preselection can induce some bias and some cell types will be difficult to characterize a posteriori. The NOS monocyte group is characterized by a low expression of CD14 and FcγRIII and the negativity of all the other markers included in the monocyte panel. This observation is in line with previous

reports from other teams. For instance, Peter Sorger's team (Harvard Medical School, Boston, USA), one of the biggest authorities in protein multiplexing worldwide, observed similar fractions of blanks (even with large panels) because some cells are negative for all the antibody markers or/and have morphologies that are difficult to segment (Lin et al, bioRxiv 2022, doi 10.1101/2022.09.28.509927).

10. Supp Fig 3B – would suggest eliminating podocytes from the dot plot since there is only 1 podocyte in that analysis.

We agree that in this analysis, the number of podocytes is low. Yet, we suggest keeping all data presentation as is, as we have no arguments indicating that this single podocyte expresses donor-derived genes is scientifically aberrant (Supp Fig 3B). This is also concordant with the results depicted in Supp Fig 3A. We agree to not emphasize this result and remain cautious about the interpretation. Therefore, we don't highlight this in the results, discussion, or conclusions. If the editor and reviewer consider it essential, we are obviously able to remove the podocytes from the analysis.

11. Fig 9 – why are there so few endothelial-immune cell interactions given the findings of vascular inflammation in ABMR?

This is an important remark. It is correct that the number of ligand/receptors pairs corresponding to endothelial-immune cell interactions seems limited in Figure 9 but it is mainly due to the representation of each subtype of HLA-related interactions which exercise a multiplier effect and minimize other interactions. Figure 9b color legend characterizes the interaction probabilities. Excluding HLA-related interactions, the strongest interaction probabilities (yellow dots) are exclusively related to endothelial cells especially through ITGB2-ICAM2 and PECAM1-PECAM1 interactions. A second explanation could be related to the type of FcγRIII+ cells-endothelium interactions which can also be driven by the recognition of the Fc fragment of an antibody such as HLA-DSA bound to the endothelium. Of course, this "third party" mechanism, considered crucial in ABMR, cannot be appreciated by gene expression-based technologies and does not appear in figure 9b. Consequently, we tackled this obstacle by running a compartmentalization analysis in figure 5, which shows that FcγRIII+ cells proportion increases in the proximity of glomeruli especially in ABMR or Mixed rejection contexts.

Reviewer #2 (expertise in nephrology, Fc receptors in kidney):

The manuscript by Lamarthée et al reports a central involvement of innate immune cells, notably FcγRIII+ NK and FcγRIII+ nonclassical monocytes, in the pathogenesis of allograft rejection suggesting several potential therapeutic targets to improve allografts. They use droplet-based single-cell RNA sequencing and described 38 markers new markers on 18 independent biopsy slides. Their data strongly pointed for the role of FcγRIII+ NK and FcγRIII+ nonclassical monocytes in antibody-mediated rejection, with specificity to the glomerular area.

Comments:

As discussed by the authors I agree that 'the relation between the histological/molecular phenotype and cellular mix in the biopsies, and the underlying causal pathways of allorecognition or even autoreactivity was not assessed' and will need further study.

Nevertheless, to obtain indirect confirmation such hypothesis, the authors may address with in vitro experiments using **blood NK and monocytes from those patients** to show antibody-mediated

cytotoxicity (ADCC or ADC). This will support the proposed mechanisms in allograft rejection.

We thank the reviewer for the positive feedback. We did not collect blood NK and monocytes from the patients included in our cohorts, so *in vitro* experiments with such patient material is not possible. In order to investigate the role of the innate cells in allograft rejection, several reports have already addressed the potential mechanisms *in vitro* using healthy volunteers' blood cells. Evidence that both NK cells and monocytes are able to perform ADCC or CDC is accumulating. Among them, Shin and colleagues (Shin et al, Transplantation 2014, PMID: 24342979) showed using an *in vitro* model of ADCC that steroids and CNIs were the most effective agents in reducing NK+ cell IFN release. In addition, a recent report suggested that murine inflammatory monocytes could be targeted using immune-modifying nanoparticles to prevent acute kidney allograft rejection (Lai et al, Kidney International 2022, PMID 35850291).

We added reference to this literature to the discussion section:

Discussion Page 24 Lines 705-713: *“Using an in vitro model of ADCC, Shin et al. showed that steroids and calcineurin inhibitors were the most effective agents in reducing IFN-g production by NK cells compared to mTOR inhibitor⁷¹ but their impact on FcγR11a+ monocytes remain to be addressed. In addition to mTOR inhibitors, fostamatinib inhibits FcγR-triggered, Syk-dependent activation⁷² and, as such, could represent a potential therapy targeting a previously unexplored mechanism of kidney rejection pathogenesis as confirmed by our in vitro results. Also daratumumab, targeting CD38 and depleting NK cells and monocytes, could be considered for treating ABMR, as suggested by Doberer et al⁷³. In addition, a recent report suggested that murine inflammatory monocytes could be targeted using immune-modifying nanoparticles to prevent acute kidney allograft rejection⁷⁴.”*

Beyond this literature, and to meet the specific request of this reviewer, we further investigated the underlying causal pathways of allorecognition. In a new CellChat analysis, we assessed both secreted signaling using FCGR3A+ NK and FCGR3A+ monocytes as source cells. As depicted in new Fig.10d, very limited cellular communications were detected in PT cells clusters which is in line with Fig.10e confirming that no significant interaction was detected in the tubules. One could speculate that tubular injury is indirect and mainly results from a metabolic shutdown associated with the vascular rejection. In contrast, FCGR3A+ NK and FCGR3A+ monocytes mainly interact with endothelial cells. More specifically, VCAM1+ EC (EC_Injured) overexpress ACKR1 and TNFRSF1A making them potential targets of monocytes-derived CXCL10 and TNF but also of NK-derived CCL5.

We added this paragraph to the results section:

Results Page 21 Lines 598-612: *“To evaluate epithelial/endothelial-immune cell communications to a greater extent, we reintegrated PT cells, endothelial cells, podocytes and immune cells of interest: FCGR3A+ NK and FCGR3A+ myeloid cells (Fig.10a). We did not identify any injury cluster of podocytes given their limited number. However, four different clusters were distinguished within PT cells corresponding to the 3 segments already described from kidney cortical area to medulla (PT_S1, PT_S2 and PT-S3 respectively, Fig.10a-c). Interestingly, a cluster corresponding to injured tubular cells expressing HAVCR1 and VCAM1 (Fig.10b) was detected⁵⁶. Regarding endothelial cells, in addition to the previously described clusters (EC_{ptc}, EC_g and EC_{vr}), we detected a cluster expressing high levels of VCAM1, suggesting an activation as previously reported⁴⁷. We performed CellChat analysis assessing both secreted signaling and cell-cell contact using FCGR3A+ NK and FCGR3A+ monocytes as source cells. Very limited cellular communications were detected in PT cells clusters*

(Fig. 10d), which is in line with Fig. 10e confirming that no significant interaction was detected between the tubules and FCGR3A+ immune cells. One could speculate that tubular injury is indirect and mainly results from a metabolic shutdown associated with the vascular rejection. In contrast, secreted signaling unraveled that FCGR3A+ monocytes mainly interact with NK cells through LGALS9 coding

for galectin 9 inferred secretion (Fig.10e).”

e.

Finally, we recently showed that anti-HLA DSA triggers such cytokine production in kidney transplant recipients' serum independent of histological lesions. This secretion could be induced by FcγRIII+ cells in an *in vitro* model of ADCC mimicking anti-HLA DSA binding on endothelium (Van Loon et al, Front Immunol, 2022, 10.3389/fimmu.2022.818569). We now expanded the exact same *in vitro* model, and show in a new figure S7 that Syk inhibition by R406, the active form of fostamatinib could reduce this secreted signaling using cells from 4 individuals. In addition, we assessed the activation of NK cells by monocytes in ABMR context *in vitro*. FCGR3A+ monocytes express LGALS9 coding for galectin 9 that could trigger NK activation. We now show that DSA recognition by monocyte can induce galectin 9 secretion by monocytes and thus NK activation through Tim3 (new Figure S7). Interestingly, Tim3 expression in NK cells was higher in rejection context and more particularly when the NK cells are closely located near non classical FcγRIII+ monocytes.

We added this paragraph to the results section:

Results Pages 22 Lines 612-630: *“We previously showed that anti-HLA DSA triggers chemokine and cytokine production in kidney transplant recipients' serum, independent of histological lesions and we demonstrated that this secretion could be induced by FcγRIII+ cells in an in vitro model of ADCC mimicking anti-HLA DSA binding on endothelium²². Here, using the same in vitro model, we found that non-classical monocytes significantly secreted more galectin 9 after anti-HLA DSA recognition (Fig.S8a-d). Therefore, we measured the expression of Tim-3, the galectin 9-receptor in FcγRIII+ NK cells in kidney biopsies using multiplexed immunofluorescence (MILAN). We observed that Tim-3 expression is increased in FcγRIII+ NK cells closely located near FcγRIII+ monocytes in acute rejection context (Fig.S8e-f) suggesting that galectin-9 secreted by FcγRIII+ monocytes could trigger NK cells activation in rejection context.*

In addition, FCGR3A+ NK and FCGR3A+ monocytes interact with endothelial cells. More specifically, VCAM1+ EC (EC_Injured) overexpress ACKR1 and TNFRSF1A making them potential targets of monocytes-derived CXCL10 and TNF but also of NK-derived CCL5. Given that the recognition of a Fc fragment by the FcγRIII receptor induce Syk dependent pathways triggering cytokine secretion, we investigated whether a Syk inhibitor blocks the secreted signaling using the exact same in vitro model. The addition of R406, the active form of fostamatinib, during the NK/non-classical monocytes/GENC coculture completely abrogated the secretion of TNF, CCL5 and CXCL10 in presence of HLA DSA, suggesting that Syk inhibitors could alleviate the cytokine release of NK cells and monocytes upon ABMR (Fig.S9a-b).”

Regarding the potential role of monocytes in supporting CD8+ T cells, we have added data from coculture studies of human primary monocytes with HLA-A2 specific CD8+ T cells. We showed that monocytes could also support the proliferation and the cytotoxicity CD8+ T cells against HLA-A2^{+/+} epithelial cells (new supplementary figure 7, Results lines 585-596, Discussion lines 677-678).

Suppl Fig. 7

% Divided cells

HLA-A2 specific
CD8+ Tcells

e.

Cytotoxic Assay

f.

AUC

Reviewer #3 (expertise in nephrology, kidney transplantation):

This paper by Lamarthée et al., examined the transcriptional and spatial profiling of the kidney allograft exploring the role of FcγRIII+ innate immune cells in mediating allograft rejection in human kidney transplant biopsies. The authors performed single-cell RNA-sequencing of 35,152 transcriptomes from 16 kidney transplant biopsies with varying rejection diagnoses and intensity of rejection these were compared to data derived from biopsies without rejection and generated cell-type specific gene expression signatures. The authors identified a specific association between recipient-derived FCGR3A+ monocytes, FCGR3A+ NK cells and the severity of allograft inflammation. Activated FCGR3A+ monocytes overexpressed CD47 and lymphocyte immunoglobulin-like receptor (LILRA) genes and increased signaling pathways that initiated T cell infiltration. FCGR3A+ NK cells overexpressed FCRL3, suggesting that antibody-dependent cytotoxic activity is a central mechanism of NK cell-mediated graft injury. Multiplexed immunofluorescence using 38 markers on 18 independent biopsy slides confirmed this role of FcγRIII+ NK and FcγRIII+ nonclassical monocytes in antibody-mediated rejection, with specificity to the glomerular area.

Critique: Overall, this is a well written and impressive manuscript that will likely change the way that clinicians look at well-established paradigms of allograft rejection, which focus primarily on adaptive immunity and T-cell activation. These investigators show that the primary initiating events and propagation are associated with innate immunity involving FcγR3A+ NK cells and FcγR3A+ monocytes. This is true for both antibody-mediated rejection (i.e., ADCC) and cell-mediated rejection where innate immune events mediated by LILRA molecules on FcγR3A+ monocytes show that these cells are not only capable of recognizing IgG binding to target cells in the allograft but also have the capacity to directly recognize non-self on donor cells through the LILRA pathway, initiating CD8+ T-cell activation. The same is true for FcγR3A+ NK cells where KIR (killer inhibitor receptor) can initiate direct recognition of non-self HLA and drive CD8+T cell activation.

Importantly, one must consider the implications of these findings for clinical therapeutics. The authors conclude: "We have demonstrated the importance of FcγRIII+ monocytes and FcγRIII+ NK cells, and innate immune crosstalk during rejection. We believe that this work is a major resource for understanding of the role of these cells and the innate immune system involvement in kidney transplant rejection and shows promise for developing therapeutic strategies that target these pathways." This is the essence and productive culmination of this type of work. In this regard, there have been some attempts to address this. Shin et al. (Shin BH, Ge S, Mirocha J, Karasyov A, Vo A, Jordan SC, Toyoda M. Regulation of anti-HLA antibody-dependent natural killer cell activation by immunosuppressive agents. *Transplantation*. 2014 Feb 15;97(3):294-300. doi: 10.1097/01.TP.0000438636.52085.50. PMID: 24342979.) showed using an in vitro analysis of ADCC looking at CD56+ NK cell activation that steroids and CNI were the most effective agents in reducing NK+ cell γ-IFN release in ADCC. The authors mention there may be promise for mTOR inhibitors, but in the study by Shin et al., no effect was seen. I think it would be important for the authors to also reference emerging data from the use of daratumumab (anti-CD38) which is used to treat multiple myeloma. CD38 is expressed on multiple cell types including NK cells and monocytes. Recent data from case studies (Doberer K, Kläger J, Gualdoni GA, Mayer KA, Eskandary F, Farkash EA, Agis H, Reiter T, Reindl-Schwaighofer R, Wahrmann M, Cohen G, Haslacher H, Bond G, Simonitsch-Klupp I, Halloran PF, Böhmig GA. CD38 Antibody Daratumumab for the Treatment of Chronic Active Antibody-mediated Kidney Allograft Rejection. *Transplantation*. 2021 Feb 1;105(2):451-457. doi: 10.1097/TP.0000000000003247.) showed that daratumumab can be effective in treating antibody-

mediated rejection, even when donor antibodies are not reduced. This suggests, possible depletion of NK cells and monocytes in the glomeruli capillary loops. Unfortunately, CD38 is also expressed on Treg cells and treatment can initiate CMR episodes. I think a mention of these studies would be helpful in directing discussion of possible future therapies based on their findings.

We thank the reviewer for the positive feedback and constructive suggestions.

As suggested, we added new citations such as Shin et al, Transplantation, 2014 in which the authors investigated steroid, CNI and mTOR inhibitor effects on NK cells ADCC capacities.

Discussion Page 24 Lines 703-707: *“We can thus speculate that pharmacological inhibition of the Syk or mTOR pathway could lead to proportional decrease in both FcγRIII+ monocytes and FcγRIII+ NK cell activation^{64,65}. Using an in vitro model of ADCC, Shin et al. showed that steroids and calcineurin inhibitors were the most effective agents in reducing IFN-γ production by NK cells compared to mTOR inhibitor but their impact on FcγRIII+ monocytes remain to be addressed.”*

In the revised version of the manuscript, we also include the citation of Doberer et al. on the potential use of daratumumab in chronic active ABMR. In line with reviewer’s statement, we indeed confirm CD38 expression in a broad range of immune cells as illustrated by the figure below:

CD38 expression was higher in monocytes and NK cells compared to CD4 T cells which could suggest that Tregs could be less impacted by a lower dose of daratumumab. Of course, this hypothesis needs to be investigated, but we consider this beyond the scope of our manuscript.

We added this to the Discussion section Page 25 Lines 710-713: *“Also daratumumab, targeting CD38 and depleting NK cells and monocytes, could be considered for treating ABMR, as suggested by Doberer et al⁷¹.”*

REVIEWER COMMENTS

Reviewer #1 (expert in spatial transcriptomics, kidney transcriptomics):

This is a resubmission of the manuscript by Lamarthee et al. entitled Transcriptional and spatial profiling of the kidney allograft unravels a central role for FcyRIII+ innate immune cells in rejection. The authors were mostly responsive to the critiques.

The improvements in the manuscript include the following:

1. Improved analysis of cell-cell interactions between immune and endothelial and PT cells in the new Figure 10. Annotation of an injury PT cluster is provided.
2. A web-based tool to view Milan images. The importance of this asset cannot be understated.
3. Validation with an external scRNAseq dataset in Supplemental Figure 6. This dataset appears to be of high quality with expected expression of canonical markers and broad cell type recovery.
4. The unsupervised rejection clusters were clarified. Thank you.

Overall, this is an exciting manuscript; however, critiques remain and include the following:

1. Framing of the key finding – In lines 109-110: “These results uncovered the central involvement of innate immune cells in the pathogenesis of allograft rejection and indicate several potential therapeutic targets to improve allograft longevity.” Given the quantity of prior work linking innate immune cells to ABMR, perhaps “uncovered” is too strong and “support” or “reinforce” or “expand upon” may be appropriate.
2. MILAN methods (line 263-264) –Consider adding a qualifier of “immune” to the sentence: The antibody panel for MILAN was designed to allow a phenotypic identification of the most abundant “immune” cell types based on the results from the scRNAseq.” Epithelial markers for most cells were not included and there is an inability to distinguish endothelial cell sub-types in the MILAN panel.
3. MILAN clarifications (Figure 3c and 3f) – How were the AQP1- PT cells defined? How were distal tubule cells defined without a distal marker? Please kindly provide the significance of the CD138 tubular cells for the reader (are these also PT cells, is it possible to have plasma cell infiltrate in ABMR)? Were some tubules defined by autofluorescence? If so, what was the pathologist’s approach to annotating injured PTs and distal tubules (i.e. dilated, epithelial simplification, etc)? Were glomeruli defined by the pathologist using autofluorescence? If so, please clarify these points in the methods.
4. Please kindly include QC measures. For scRNAseq, what proportion of cells were excluded per sample based on mitochondrial reads or low gene number? For Milan, what proportion of CD4+ cells co-labelled with CD3?
5. The authors were unable to expand the sample size. As a result, the limitations section (starting line 687) could be enhanced by clearly identifying the following:
 - a. The scRNAseq sample size was insufficient to identify common cell types of the kidney such as the distal convoluted tubule or principal cell. Few podocytes were obtained.
 - b. Ubiquitous stress marker expression of MIOX in the PT, and SPP1 (with low EGF) in the LOH provide a clue that the wet lab preparation technique for these samples was not fully optimized.
 - i. As an aside, a poor sample prep could explain the paucity of non-immune cell types recovered. An alternative interpretation is that these samples had diffuse biological cellular injury, but such diffuse injury was not described by the authors or apparent in the images provided.
 - c. Marker selection for the MILAN assay focused on immune cell sub-types, largely excluding common epithelial cell markers (e.g. SLC12A1 for LOH) and injury markers. As a result, epithelial-immune neighborhoods were unable to be fully assessed.
 - d. A non-overlapping set of samples were used in scRNAseq and MILAN.

Reviewer #2 (expert in nephrology, Fc receptors in kidney):

The authors replied satisfactorily to my comments including new data and sufficient results from literature to answer to my questions. I have no additional comments.

Reviewer #3 (expert in nephrology, kidney transplantation):

This paper by Naesens represents an extensively revised submission which I feel has appropriately addressed the concerns of the reviewers. I think the manuscript is very comprehensive, well written and of great importance to the transplant community. I feel it is now acceptable in the revised form.

RESPONSE TO REVIEWERS' COMMENTS

Reviewer #1 (expert in spatial transcriptomics, kidney transcriptomics):

This is a resubmission of the manuscript by Lamarthee et al. entitled Transcriptional and spatial profiling of the kidney allograft unravels a central role for FcyRIII+ innate immune cells in rejection. The authors were mostly responsive to the critiques.

The improvements in the manuscript include the following:

1. Improved analysis of cell-cell interactions between immune and endothelial and PT cells in the new Figure 10. Annotation of an injury PT cluster is provided.
2. A web-based tool to view Milan images. The importance of this asset cannot be understated.
3. Validation with an external scRNAseq dataset in Supplemental Figure 6. This dataset appears to be of high quality with expected expression of canonical markers and broad cell type recovery.
4. The unsupervised rejection clusters were clarified. Thank you.

We thank the reviewer for the positive feedback.

Overall, this is an exciting manuscript; however, critiques remain and include the following:

1. Framing of the key finding – In lines 109-110: “These results uncovered the central involvement of innate immune cells in the pathogenesis of allograft rejection and indicate several potential therapeutic targets to improve allograft longevity.” Given the quantity of prior work linking innate immune cells to ABMR, perhaps “uncovered” is too strong and “support” or “reinforce” or “expand upon” may be appropriate.

We changed “uncovered” for “expand upon”.

2. MILAN methods (line 263-264) –Consider adding a qualifier of “immune” to the sentence: The antibody panel for MILAN was designed to allow a phenotypic identification of the most abundant “immune” cell types based on the results from the scRNAseq.” Epithelial markers for most cells were not included and there is an inability to distinguish endothelial cell subtypes in the MILAN panel.

We added “immune” to the indicated sentence.

3. MILAN clarifications (Figure 3c and 3f) – How were the AQP1- PT cells defined? How were distal tubule cells defined without a distal marker? Please kindly provide the significance of the CD138 tubular cells for the reader (are these also PT cells, is it possible to have plasma cell infiltrate in ABMR)? Were some tubules defined by autofluorescence? If so, what was the pathologist’s approach to annotating injured PTs and distal tubules (i.e. dilated, epithelial simplification, etc)? Were glomeruli defined by the pathologist using autofluorescence? If so, please clarify these points in the methods.

At transcriptomic level, *AQP1* and *SDC1* (encoding CD138) are highly expressed in PT cells in both our dataset but also in native kidney as describe in The Human Protein Atlas:

(<https://www.proteinatlas.org/ENSG00000240583-AQP1/single+cell+type/kidney>,
<https://www.proteinatlas.org/ENSG00000115884-SDC1/single+cell+type/kidney>).

AQP1 expression is higher in PT cells as compared to Collecting duct and Distal tubular cells whereas *SDC1* level seems equivalent in cortical and medullar area. This *SDC1* gradient in PT cells is currently explored in another manuscript dedicated to structural cells 'role in kidney transplant rejection.

In MILAN, we used a combination of CD138, AQP1, PanCK and autofluorescence to segment the tubular compartment of the nephron and CD31 to identify the glomeruli.

Glomeruli were manually annotated by an experienced pathologist (FB) combining morphological recognition and a CD31 staining highlighting blood vessels.

We clarified this in the updated methods section:

“Glomerular regions and large vessels were manually annotated by an experienced pathologist (FB) on the autofluorescent images.”

Regarding the tubular compartment, we adopted a two-step approach. We first identified the proximal tubuli as PanCK+S100+AQP1-CD138- and the distal tubuli as PanCK-S100-AQP1+CD138+. In addition, we observed a portion of PanCK+AQP1+ tubuli as PanCK+S100-AQP1+CD138- and we indeed identified some CD138+ tubuli that have an overlapping phenotype with plasma cells. Despite the fact that we did not have a marker for loop of henle cells, LOH cells and CD138+ tubuli are easily recognizable morphologically and differ from plasma cells. Therefore, FB trained a pixel classifier using QuPath based on a composite image obtained by fusing PanCK, S100, AQP1, CD138 staining with autofluorescence for morphological information. Once the trained classifier reached the optimal level of separation between the tubular and the interstitial compartment, as assessed in a supervised manner by the pathologist, the annotations were used to distinguish plasma cells (located in the interstitium, but generally very few plasma cells were found in the infiltrate) and CD138+ tubuli.

4. Please kindly include QC measures. For scRNAseq, what proportion of cells were excluded per sample based on mitochondrial reads or low gene number? For Milan, what proportion of CD4+ cells co-labelled with CD3?

In the scRNAseq dataset, $32.5 \pm 7.5\%$ of cells on average were excluded based on mitochondrial reads or low gene number. Please find below the QC measures:

QC Measures Genes. vs UMIs per Cell

Biopsy #1

Biopsy #2

Biopsy #3

Biopsy #4

Biopsy #5

Biopsy #6

Excluded

Biopsy #7

Biopsy #8

Biopsy #9

Biopsy #10

Biopsy #11

Biopsy #12

Biopsy #13

Biopsy #14

Biopsy #15

Biopsy #16

mitochondrial reads
cut-off = 0.25

Genes

UMIs per cell

5. The authors were unable to expand the sample size. As a result, the limitations section (starting line 687) could be enhanced by clearly identifying the following:

a. The scRNAseq sample size was insufficient to identify common cell types of the kidney such as the distal convoluted tubule or principal cell. Few podocytes were obtained.

We are aware of the technical limitations of scRNASeq and the analytical pipelines, as we have recently reviewed in detail: Deleersnijder et al. JASN 2021 doi: 10.1681/ASN.2021020157. Sample size is perhaps not the main issue. The way biopsies are harvested (cortical vs medular zone), the wet lab preparation and the unbiased clustering all explain differences between data sets.

We already referred to this in the limitations section: *“The choice of technique for single cell dissociation affects the cellular mix and transcriptional profiles of the cells (REF Deleersnijder et al).”*

We have adapted this sentence in the discussion section to expand on this important point: *“The sample size of our biopsy cohorts, the way biopsies are obtained (primarily cortical tissue), the choice of technique for single cell dissociation and wet lab preparation, and the downstream clustering affects the cellular mix and transcriptional profiles of the cells both in scRNASeq (REF Deleersnijder et al). and in MILAN, and explain why in the scRNASeq dataset no distal convoluted tubule or principal cell and only few podocytes were obtained.”*

b. Ubiquitous stress marker expression of MIOX in the PT, and SPP1 (with low EGF) in the LOH provide a clue that the wet lab preparation technique for these samples was not fully optimized. As an aside, a poor sample prep could explain the paucity of non-immune cell types recovered. An alternative interpretation is that these samples had diffuse biological cellular injury, but such diffuse injury was not described by the authors or apparent in the images provided.

We agree that the specificities of the wet lab preparation techniques can impact the cell composition and expression patterns. We followed a very similar wet lab preparation technique as described for human kidney biopsies in Rashmi et al (Am J Transplant 2022 doi <https://doi.org/10.1111/ajt.16871>) or Suryawanshi et al (Plos One 2022 <https://doi.org/10.1371/journal.pone.0267704>) or Malone et al (J Am Soc Nephrol 2020 10.1681/ASN.2020030326) or Wu et al (J Am Soc Nephrol 2018 10.1681/ASN.2018020125) with a mechanical dissociation followed by enzymatic digestion, wash, filtering and resuspension of the cells in PBS-BSA. We used a different enzyme (Collagenase vs Liberase), but to our knowledge this enzyme showed superior efficacy to digest kidney tissue to get single cell suspension from human kidney biopsies as an enzymatic mix containing Collagenase + DNase I + Hyaluronidase presented superior results as compared to Liberase treatment (Yaigoub et al, Front Cell Dev Biol 2022 doi: 10.3389/fcell.2022.822275).

We have expanded on this in the limitations section of the revised manuscript (see above): *“The sample size of our biopsy cohorts, the way biopsies are obtained (primarily cortical tissue), the choice of technique for single cell dissociation and wet lab preparation, and the downstream clustering affects the cellular mix and*

transcriptional profiles of the cells both in scRNASeq (REF Deleersnijder et al). and in MILAN, and explain why in the scRNASeq dataset no distal convoluted tubule or principal cell and only few podocytes were obtained.”

c. Marker selection for the MILAN assay focused on immune cell sub-types, largely excluding common epithelial cell markers (e.g. SLC12A1 for LOH) and injury markers. As a result, epithelial-immune neighborhoods were unable to be fully assessed.

We agree with the reviewer that we did not perform neighborhood analysis with precise epithelial subtypes especially with medulla-related epithelial cells such as LOH. It is possible that neighborhood analysis between immune cells and medulla might be relevant but to our knowledge, the rejection and more particularly in ABMR context is mainly driven in the cortical zone in the endothelial compartment. We added a sentence in the discussion section:

“Moreover, in MILAN-derived dataset, we largely focused on immune cell subtypes, excluding common epithelial cell markers especially from the medulla area (e.g. SLC12A1 for LOH) and injury markers. As a result, epithelial-immune neighborhoods were not fully assessed at epithelial subtypes level.”

d. A non-overlapping set of samples were used in scRNAseq and MILAN.

We have now added a sentence in the discussion regarding this issue: *“In addition, we used a non-overlapping set of samples in scRNAseq and MILAN.”*

Reviewer #2 (expert in nephrology, Fc receptors in kidney):

The authors replied satisfactorily to my comments including new data and sufficient results from literature to answer to my questions. I have no additional comments.

We thank the reviewer for the positive feedback.

Reviewer #3 (expert in nephrology, kidney transplantation):

This paper by Naesens represents an extensively revised submission which I feel has appropriately addressed the concerns of the reviewers. I think the manuscript is very comprehensive, well written and of great importance to the transplant community. I feel it is now acceptable in the revised form.

We thank the reviewer for the positive feedback.

REVIEWERS' COMMENTS

Reviewer #1 (Remarks to the Author):

The authors have addressed all of the critiques. The additional clarity in the limitations is appreciated. I wish them good fortune with their future scientific undertakings.